# Communication-efficient SGD: From Local SGD to One-Shot Averaging

**Artin Spiridonoff**
Division of Systems Engineering
Boston University
Boston, MA 02215
artin@bu.edu

**Alex Olshevsky**
Division of Systems Engineering
Boston University
Boston, MA 02215
alexols@bu.edu

**Ioannis Ch. Paschalidis**
Division of Systems Engineering
Boston University
Boston, MA 02215
yannisp@bu.edu

## Abstract

We consider speeding up stochastic gradient descent (SGD) by parallelizing it across multiple workers. We assume the same data set is shared among $N$ workers, who can take SGD steps and coordinate with a central server. While it is possible to obtain a linear reduction in the variance by averaging all the stochastic gradients at every step, this requires a lot of communication between the workers and the server, which can dramatically reduce the gains from parallelism. The Local SGD method, proposed and analyzed in the earlier literature, suggests machines should make many local steps between such communications. While the initial analysis of Local SGD showed it needs $\Omega(\sqrt{T})$ communications for $T$ local gradient steps in order for the error to scale proportionately to $1/(NT)$, this has been successively improved in a string of papers, with the state of the art requiring $\Omega\left(N\left(\text{poly}(\log T)\right)\right)$ communications. In this paper, we suggest a Local SGD scheme that communicates less overall by communicating less frequently as the number of iterations grows. Our analysis shows that this can achieve an error that scales as $1/(NT)$ with a number of communications that is completely independent of $T$. In particular, we show that $\Omega(N)$ communications are sufficient. Empirical evidence suggests this bound is close to tight as we further show that $\sqrt{N}$ or $N^{3/4}$ communications fail to achieve linear speed-up in simulations. Moreover, we show that under mild assumptions, the main of which is twice differentiability on any neighborhood of the optimal solution, one-shot averaging which only uses a single round of communication can also achieve the optimal convergence rate asymptotically.

## 1 Introduction

Stochastic Gradient Descent (SGD) is a widely used algorithm to minimize convex functions $f$ in which model parameters are updated iteratively as

$$\mathbf{x}^{t+1} = \mathbf{x}^t - \eta_t \hat{\mathbf{g}}^t,$$

where $\hat{\mathbf{g}}^t$ is a stochastic gradient of $f$ at the point $\mathbf{x}^t$ and $\eta_t$ is the learning rate. This algorithm can be naively parallelized by adding more workers independently to compute a gradient and then average

35th Conference on Neural Information Processing Systems (NeurIPS 2021).

them at each step to reduce the variance in estimation of the true gradient $\nabla f(\mathbf{x}^t)$ (Dekel et al., 2012). This method requires each worker to share their computed gradients with each other at every iteration. We will refer to this method as "synchronized parallel SGD."

However, it is widely acknowledged that communication is a major bottleneck of this method for large scale optimization applications (McMahan et al., 2017; Konečnỳ et al., 2016; Lin et al., 2018b). Often, mini-batch parallel SGD is suggested to address this issue by increasing the computation to communication ratio. Nonetheless, too large mini-batch size might degrade performance (Lin et al., 2018a). Along the same lines of increasing the computation over communication effort, *local* SGD has been proposed to reduce communications (McMahan et al., 2017; Dieuleveut, Patel, 2019). In this method, workers compute (stochastic) gradients and update their parameters locally, and communicate only once in a while to obtain the average of their parameters. Local SGD improves the communication efficiency not only by reducing the number of communication rounds, but also alleviates the synchronization delay caused by waiting for slow workers and evens out the variations in workers' computing time (Wang, Joshi, 2018b).

On the other hand, since individual gradients of each worker are calculated at different points, this method introduces residual error as opposed to fully synchronized SGD. Therefore, there is a trade-off between having fewer communication rounds and introducing additional errors to the gradient estimates.

The idea of making local updates is not new and has been used in practice for a while (Mangasarian, 1995; Konečnỳ et al., 2016). However, until recently, there have been few successful efforts to analyze Local SGD theoretically and therefore it is not fully understood yet. Zhang et al. (2016) show that for quadratic functions, when the variance of the noise is higher far from the optimum, frequent averaging leads to faster convergence. The first question we try to answer in this work is: how many communication rounds are needed for Local SGD to have the *similar* convergence rate of a synchronized parallel SGD while achieving performance that linearly improves in the number of workers?

Stich (2019) was among the first who sought to answer this question for general strongly convex and smooth functions and showed that the communication rounds can be reduced up to a factor of $H = \mathcal{O}(\sqrt{T/N})$, without affecting the asymptotic convergence rate (up to constant factors), where $T$ is the total number of iterations and $N$ is number of parallel workers.

Focusing on smooth and possibly non-convex functions which satisfy a Polyak-Lojasiewicz condition, Haddadpour et al. (2019) demonstrate that only $R = \Omega((TN)^{1/3})$ communication rounds are sufficient to achieve asymptotic performance that scales proportionately to $1/N$.

Recently, Khaled et al. (2020) and Stich, Karimireddy (2019) improve upon the previous works by showing linear speed-up for Local SGD with only $\Omega(N \text{ poly log }(T))$ communication rounds when data is identically distributed among workers and $f$ is strongly convex. Their works also consider the cases when $f$ is not necessarily strongly convex as well as the case of data being heterogeneously distributed among workers.

More recently, Yuan, Ma (2020) proposed a new *accelerated* method that requires only $\Omega(N^{1/3} \text{ poly log }(T))$ communication rounds for linear speed-up. While their results improve upon the earlier work, the communication requirements remain dependent on the total iterations $T$.

One-Shot Averaging (OSA), a method that takes an extreme approach to reducing communication, involves workers performing local updates until the very end when they average their parameters (Mcdonald et al., 2009; Zinkevich et al., 2010; Zhang et al., 2013c; Rosenblatt, Nadler, 2016; Godichon-Baggioni, Saadane, 2020). This method can be seen as an extreme case of Local SGD with $R = 1$ and $H = T$ local steps. Dieuleveut, Patel (2019); Godichon-Baggioni, Saadane (2020) provide an analysis of OSA and show that asymptotically, linear speed-up in the number of workers is achieved for a weighted average of iterates. However, both of these works make restrictive assumptions such as uniformly three-times continuously differentiability and bounded second and third derivatives or twice differentiability almost everywhere with bounded Hessian, respectively. The second question we attempt to answer in this work, is whether these assumptions can be relaxed and OSA can achieve linear speed-up in more general scenarios.

In this work, we focus on smooth and strongly convex functions with a general noise model. Our contributions are three-fold:

Table 1: Comparison of Similar Works

| Reference | Convergence rate $f(\hat{\mathbf{x}}^T) - f^{*}$ [a] | Communication Rounds $R$ | Noise model |
|---|---|---|---|
| Stich (2019) | $\mathcal{O}\left(\frac{\xi^0}{R^3} + \frac{\sigma^2}{\mu NT} + \frac{\kappa G^2}{\mu R^2}\right)$ [b] | $\Omega(\sqrt{TN})$ | uniform |
| Haddadpour et al. (2019) | $\mathcal{O}\left(\frac{\xi^0}{R^3} + \frac{\kappa\sigma^2}{\mu NT} + \frac{\kappa^2\sigma^2}{\mu NTR}\right)$ | $\Omega((TN)^{1/3})$ | uniform with strong-growth [c] |
| Stich, Karimireddy (2019) | $\tilde{\mathcal{O}}\left(\text{exp. decay} + \frac{\sigma^2}{\mu NT}\right)$ [d] | $\Omega(N * \text{poly}(\log T))$ | uniform with strong-growth |
| Woodworth et al. (2020) | $\mathcal{O}\left(\text{exp. decay} + \frac{\sigma^2}{\mu NT} + \frac{\kappa\sigma^2 \log(9+T/\kappa)}{\mu TR}\right)$ | $\Omega(N * \text{poly}(\log T))$ | uniform |
| Khaled et al. (2020) | $\tilde{\mathcal{O}}\left(\frac{\kappa\xi^0}{T^2} + \frac{\kappa\sigma^2}{\mu NT} + \frac{\kappa^2\sigma^2}{\mu TR}\right)$ | $\Omega(N * \text{poly}(\log T))$ | uniform |
| Yuan, Ma (2020) | $\tilde{\mathcal{O}}\left(\text{exp. decay} + \frac{\sigma^2}{\mu NT} + \frac{\kappa^2\sigma^2}{\mu TR^3}\right)$ [e] | $\Omega(N^{1/3} * \text{poly}(\log T))$ | uniform |
| **This Paper** | $\mathcal{O}\left(\frac{(1+c\kappa^2 \ln(TR^{-2}))\xi^0}{\kappa^{-2}T^2} + \frac{\kappa\sigma^2}{\mu NT} + \frac{\kappa^2\sigma^2}{\mu TR}\right)$ [f] | $\Omega(N)$ | uniform with strong-growth |

[a] Depending on the work, $\hat{\mathbf{x}}^T$ is either the last iterate or a weighted average of iterates up to $T$.

[b] $G$ is the uniform upper bound assumed for the $l_2$ norm of gradients in the corresponding work.

[c] This noise model is defined in Assumption 5.

[d] $\tilde{\mathcal{O}}(.)$ ignores the poly-logarithmic and constant factors.

[e] This is the bound for FedAC-II. FedAC-I requires $R = \Omega(N^{1/2} * \text{poly}(\log T))$.

[f] $c$ is the multiplicative factor in the noise model defined in Assumption 5.

1. We propose a communication strategy which requires only $R = \Omega(N)$ communication rounds to achieve performance that scales as $1/N$ in the number of workers. To the best of the authors' knowledge, this is the only work to show that the number of communications can be taken to be completely independent of $T$. All previous papers required a number of communications which was at least $N$ times a polynomial in $\log(T)$, or had a stronger scaling with $T$. A comparison of our result to the available literature can be found in Table 1.

2. We show under mild additional assumptions, in particular twice differentiability on a neighborhood of the optimal point, OSA reaches linear speed-up asymptotically, i.e., with only one communication round we achieve the convergence rate of $\mathcal{O}(1/(NT))$.

3. We simulate a simple example which is not twice differentiable at the optimum and observe that our bounds for part 1. are reasonably close to being tight. In particular, using 1 or $\sqrt{N}$ or $N^{3/4}$ communications does not appear to result in a linear speed-up in the number of workers (while $N$ communications does give a linear speed-up).

We notice that FedAC (Yuan, Ma, 2020) has a better dependence on the number of workers $N$, in expense of (poly logarithmic) dependence on $T$. With that in mind, we still believe our communication strategy is of independent interest, particularly in the framework of non-accelerated methods. We have performed extensive numerical experiments and comparisons between the two methods and highlighted the regimes where each method outperforms the other.

It is worth mentioning that although the the communication complexity by Woodworth et al. (2020) depends on $T$, their bound has a lower dependence on condition number $\kappa$. Hence, their results are stronger than ours only when $\kappa = \Omega(\log T)$.

The rest of this paper is organized as follows. In the following subsection we outline the related literature and ongoing works. In Section 2 we define the main problem and state our assumptions. We present our theoretical findings in Section 3 followed by numerical experiments in Section 4 and conclusion remarks in Section 5.

## 1.1 Related work

There has been a lot of effort in the recent research to take into account the communication delays and training time in designing faster algorithms (McDonald et al., 2010; Zhang et al., 2015; Bijral et al., 2016; Kairouz et al., 2019). See (Tang et al., 2020) for a comprehensive survey of commu-

nication efficient distributed training algorithms considering both system-level and algorithm-level optimizations.

Many works study the communication complexity of distributed methods for convex optimization (Arjevani, Shamir, 2015; Woodworth et al., 2020) and statistical estimation (Zhang et al., 2013b). Woodworth et al. (2020) present a rigorous comparison of Local SGD with $H$ local steps and mini-batch SGD with $H$ times larger mini-batch size and the same number of communication rounds (we will refer to such a method as large mini-batch SGD) and show regimes in which each algorithm performs better: they show that Local SGD is strictly better than large mini-batch SGD when the functions are quadratic. Moreover, they prove a lower bound on the worst case of Local SGD that is higher than the worst-case error of large mini-batch SGD in a certain regime. Zhang et al. (2013b) study the minimum amount of communication required to achieve centralized minimax-optimal rates by establishing lower bounds on minimax risks for distributed statistical estimation under a communication budget.

A parallel line of work studies the convergence of Local SGD with non-convex functions Zhou, Cong (2018). Yu et al. (2019) was among the first works to present provable guarantees of Local SGD with linear speed-up. Wang, Joshi (2018b) and Koloskova et al. (2020) present unified frameworks for analyzing decentralized SGD with local updates, elastic averaging or changing topology. The follow-up work of Wang, Joshi (2018a) presents ADACOMM, an adaptive communication strategy that starts with infrequent averaging and then increases the communication frequency in order to achieve a low error floor. They analyze the error-runtime trade-off of Local SGD with nonconvex functions and propose communication times to achieve faster runtime.

Another line of work reduces the communication by compressing the gradients and hence limiting the number of bits transmitted in every message between workers (Lin et al., 2018b; Alistarh et al., 2017; Wangni et al., 2018; Stich et al., 2018; Stich, Karimireddy, 2019).

Asynchronous methods have been studied widely due to their advantages over synchronized methods which suffer from synchronization delays due to the slower workers (Spiridonoff et al., 2020). Wang et al. (2019) study the error-runtime trade-off in decentralized optimization and proposes MATCHA, an algorithm which parallelizes inter-node communication by decomposing the topology into matchings. However, these methods are relatively more involved and they often require full knowledge of the network, solving a semi-definite program and/or calculating communication probabilities (schedules) as in Hendrikx et al. (2019).

**The homogeneous data assumption.**    In this work, we focus on the case when the data distribution is the same across workers. A number of previous works (Khaled et al., 2020; Haddadpour et al., 2019; Stich, 2019; Dieuleveut, Patel, 2019) studied local SGD under this assumption. The assumption is valid when the same data set is either shared across multiple workers in the same cluster, or the assignment of data points to workers is random so that any distributional differences are small. Sharing the data set across multiple workers in this way is a popular strategy to speed up training. For example, such data sharing is implemented in (Chen et al., 2012; Yadan et al., 2013; Zhang et al., 2013a) to speed up training of deep neural networks with multiple GPUs within a single server. While there are many widely used mechanisms such as Horovod (Sergeev, Del Balso, 2018) for *synchronous* data-parallel distributed training, they share a major communication bottleneck of broadcasting gradients to all workers (Grubic et al., 2018). Local SGD improves on these methods by reducing the communication of model parameters from every iteration to a smaller number of rounds during the entire optimization process. Our approach further reduces the communication overhead by communicating less as the number of iterations grows.

## 1.2   Notation

For a positive integer $s$, we define $[s] := \{1, \ldots, s\}$. We use bold letters to represent vectors. We denote vectors of all $0$s and $1$s by $\mathbf{0}$ and $\mathbf{1}$, respectively. We use $\|\cdot\|$ for the Euclidean norm of a vector and spectral norm of a matrix. Finally, $\mathcal{N}(\mu, \sigma^2)$ denotes a normal distribution with mean $\mu$ and variance $\sigma^2$.

## 2 Problem formulation

Suppose there are $N$ workers $\mathcal{V} = \{1, \ldots, N\}$, trying to minimize $f : \mathbb{R}^d \to \mathbb{R}$ in parallel. We assume all workers have access to $f$ through noisy gradients. In Local SGD, workers perform local gradient steps and occasionally calculate the average of all workers' iterates. Each worker $i$ holds a local parameter $\mathbf{x}_i^t$ at iteration $t$. There is a set $\mathcal{I} \subset [T]$ of communication times and nodes perform the following update:

$$\mathbf{x}_i^{t+1} = \begin{cases} x_i^t - \eta_t \hat{\mathbf{g}}_i^t, & \text{if } t+1 \notin \mathcal{I}, \\ \frac{1}{N} \sum_{j=1}^{N} (\mathbf{x}_j^t - \eta_t \hat{\mathbf{g}}_j^t), & \text{if } t+1 \in \mathcal{I}, \end{cases} \tag{1}$$

where $\hat{\mathbf{g}}_i^t$ is an unbiased stochastic gradient of $f$ at $\mathbf{x}_i^t$. When $\mathcal{I} = [T]$, we recover fully synchronized parallel SGD while $\mathcal{I} = \{T\}$ recovers one-shot averaging. Pseudo-code for Local SGD is provided as Algorithm 1.

---

**Algorithm 1** Local SGD

---

1: Input: $\mathbf{x}_i^0 = \mathbf{x}^0$ for all $i \in [n]$, total number of iterations $T$, the step-size sequence $\{\eta_t\}_{t=0}^{T-1}$, and $\mathcal{I} \subseteq [T]$
2: **for** $t = 0, \ldots, T-1$ **do**
3:    **for** $j = 1, \ldots, N$ **do**
4:       evaluate a stochastic gradient $\hat{\mathbf{g}}_j^t$
5:       **if** $t+1 \in \mathcal{I}$ **then**
6:          $\mathbf{x}_j^{t+1} = \frac{1}{N} \sum_{i=1}^{N} (\mathbf{x}_i^t - \eta_t \hat{\mathbf{g}}_i^t)$
7:       **else**
8:          $\mathbf{x}_j^{t+1} = \mathbf{x}_j^t - \eta_t \hat{\mathbf{g}}_j^t$
9:       **end if**
10:    **end for**
11: **end for**

---

Next we state the assumptions that we will use in our results. Note that we will not require all of them to hold at once.

**Assumption 1** (smoothness). *The function $f : \mathbb{R}^d \to \mathbb{R}$ is continuously differentiable and its gradients are $L$-Lipschitz, i.e.,*

$$\|\nabla f(\mathbf{x}) - \nabla f(\mathbf{y})\| \leq L \|\mathbf{x} - \mathbf{y}\|, \qquad \forall \mathbf{x}, \mathbf{y}.$$

**Assumption 2** (strong convexity). *$f$ is $\mu$-strongly convex with $\mu > 0$, i.e.,*

$$f(\mathbf{x}) + \langle \mathbf{g}, \mathbf{y} - \mathbf{x} \rangle + \frac{\mu}{2} \|\mathbf{x} - \mathbf{y}\|^2 \leq f(\mathbf{y}), \qquad \forall \mathbf{x}, \mathbf{y} \in \mathbb{R}^d, \forall \mathbf{g} \in \partial f(\mathbf{x}),$$

*where $\partial f(\mathbf{x})$ denotes the set of subgradients of $f$ at $\mathbf{x}$. When $f$ is also continuously differentiable, $\partial f(\mathbf{x}) = \{\nabla f(\mathbf{x})\}$.*

Note that when $f$ satisfies Assumption 2, it has a *unique* optimal point $\mathbf{x}^*$ where $f(\mathbf{x}^*) = f^*$ where $f^* = \min_{\mathbf{x}} f(\mathbf{x})$.

**Assumption 3** (Polyak-Łohasiewicz condition). *$f$ is $\mu$-Polyak-Łohasiewicz ($\mu$-PL for short) if*

$$\|\nabla f(\mathbf{x})\|^2 \geq 2\mu (f(\mathbf{x}) - f^*), \qquad \forall \mathbf{x}.$$

*where $f^* = \min_{\mathbf{x}} f(\mathbf{x})$ is the global minimum of $f$. We further assume that $f$ has a* unique *optimal point $\mathbf{x}^*$ where $f(\mathbf{x}^*) = f^*$.*

When $f$ satisfies both Assumptions 1 and 2 or Assumptions 1 and 3, we define $\kappa = L/\mu$ as the condition number of $f$.

Strong convexity implies the PL condition but the reverse does not always hold. For instance, the logistic regression loss function satisfies the PL condition over any compact set (Karimi et al., 2016). In fact, a PL function is not even necessarily convex. Charles, Papailiopoulos (2018) show that deep networks with linear activation functions are PL almost everywhere in the parameter space. Allen-Zhu et al. (2018) show, with high probability over random initializations, that sufficiently wide recurrent neural networks satisfy the PL condition. Therefore, the PL condition is more applicable, especially in the context of neural networks (Madden et al., 2020).

**Assumption 4** (twice differentiability at the optimum). *$f$ is twice continuously differentiable on an open set containing the optimal point $\mathbf{x}^*$.*

We make the following assumption on the noise of stochastic gradients, using $\mathbf{w}_i^t = \hat{\mathbf{g}}_i^t - \nabla f(\mathbf{x}_i^t)$ to denote the difference between the stochastic and true gradients.

**Assumption 5** (uniform with strong-growth noise). *Conditioned on the iterate $\mathbf{x}_i^t$, the random variable $\mathbf{w}_i^t$ is zero-mean and independent with its expected squared norm error bounded as,*

$$\mathbb{E}[\|\mathbf{w}_i^t\|^2|\mathbf{x}_i^t] \leq c\|\nabla f(\mathbf{x}_i^t)\|^2 + \sigma^2,$$

*where $\sigma^2, c \geq 0$ are constants.*

The noise model of Assumption 5 is very general and it includes the common case with uniformly bounded squared norm error when $c = 0$. As it is noted by Zhang et al. (2016), the advantage of periodic averaging compared to one-shot averaging only appears when $c/\sigma^2$ is large. Therefore, to study Local SGD, it is important to consider a noise model as in Assumption 5 to capture the effects of frequent averaging. Among the related works mentioned in Table 1, only Stich, Karimireddy (2019) and Haddadpour et al. (2019) analyze this noise model while the rest study the special case with $c = 0$. SGD under this noise model with $c > 0$ and $\sigma^2 = 0$ was first studied in Schmidt, Roux (2013) under the name *strong-growth condition*. Therefore we refer to the noise model considered in this work as *uniform with strong-growth*.

**Assumption 6** (sub-Gaussian noise). *Conditioned on the iterate $\mathbf{x}_i^t$, random variable $\mathbf{w}_i^t$ is zero-mean, independent and $[\mathbf{w}_i^t]_l$ is $(\sigma/\sqrt{d})$-sub-Gaussian, for $l = 1, \ldots, d$, i.e.,*

$$\mathbb{E}[\exp(\lambda([\mathbf{w}_i^t]_l - \mathbb{E}[\mathbf{w}_i^t]_l))|\mathbf{x}_i^t] \leq \exp\left(\frac{\lambda^2\sigma^2}{2d}\right), \qquad \forall \lambda \in \mathbb{R}, l = 1, \ldots, d.$$

*Thus, it has uniformly bounded variance $\mathbb{E}[\|\mathbf{w}_i^t\|^2|\mathbf{x}_i^t] \leq \sigma^2$.*

A sub-Gaussian noise model is commonly assumed for deriving concentration bounds for SGD, which we will use to prove our results for OSA.

As already mentioned in the Introduction, the main goal of this paper is to study the effect of communication times on the convergence of the Local SGD and provide better theoretical guarantees. In what follows, we claim that by carefully choosing the communication times, linear speed-up of parallel SGD can be attained with only a small number of communication instances. Moreover, we will obtain a set of sufficient conditions for OSA to achieve linear speed-up.

# 3 Convergence results

In this section we present our main convergence results for Local SGD and OSA. In what follows, we denote by $\bar{\mathbf{x}}^t := (\sum_{i=1}^N \mathbf{x}_i^t)/N$ the average of the iterates of all workers. Notice that $\mathbf{x}_i^t = \bar{\mathbf{x}}^t$ for $t \in \mathcal{I}$ and $i = 1, \ldots, N$.

## 3.1 Local SGD

Let us introduce the notation

$$0 = \tau_0 < \tau_1 < \ldots < \tau_R = T,$$

for the communication times. Further, let us define $H_i := \tau_{i+1} - \tau_i$ to be the $i$'th interc-communication interval. Our first theorem gives a performance bound under the assumption that $H_i$ grows linearly with $i$.

**Theorem 1.** *Suppose Assumptions 1 (smoothness), 2 (strong convexity) and 5 (uniform with strong growth noise) hold.*

*Choose the parameters as follows: $R$ such that $1 \leq R \leq \sqrt{2T}$ and $a := \lceil 2T/R^2 \rceil \geq 1$, $H_i = a(i+1)$ and $\tau_{i+1} = \min(\tau_i + H_i, T)$ for $i = 0, \ldots, R-1$. Choose $\beta \geq \max\{9\kappa, 12\kappa^2 c \max\{\ln(3), \ln(1 + T/(4\kappa R^2))\} + 3\kappa(1 + c/N)\}$ and set the learning rate as $\eta_t = 3/\mu(t+\beta), t = 0, 1, \ldots, T-1$.*

*Then using Algorithm 1 we have,*

$$\mathbb{E}[f(\bar{\mathbf{x}}^T)] - f^* \leq \frac{\beta^2(f(\bar{\mathbf{x}}^0) - f^*)}{T^2} + \frac{9L\sigma^2}{2\mu^2 NT} + \frac{144L^2\sigma^2}{\mu^3 RT}.$$

**Corollary 1.** *Under the assumptions of Theorem 1, selecting the number of communications $R = \Omega(\kappa N)$ we obtain*

$$\mathbb{E}[f(\bar{\mathbf{x}}^T)] - f^* \leq \frac{\beta^2(f(\bar{\mathbf{x}}^0) - f^*)}{T^2} + \mathcal{O}\left(\frac{L\sigma^2}{\mu^2 NT}\right).$$

The choice of communication times in Theorem 1 aligns with the intuition that workers need to communicate more frequently at the beginning of the optimization. As the the step-sizes become smaller and workers' local parameters get closer to the global minimum, they diverge more slowly from each other and therefore, less communication is required to re-align them. The advantage of this communication strategy over fixed periodic averaging has been only empirically shown in Haddadpour et al. (2019). The proof of Theorem 1 can be found in Appendix B.

### 3.2 One-shot averaging

The previous literature literature has shown OSA achieves asymptotic linear speed-up under some restrictive assumptions. For instance, Dieuleveut, Patel (2019) show this for three times continuously differentiable functions with second and third uniformly bounded derivatives. Similarly, Godichon-Baggioni, Saadane (2020) require the objective function to be strongly convex, twice continuously differentiable almost everywhere, with a bounded Hessian everywhere and gradients satisfying the following condition for some constant $C_m$ and all $\mathbf{x} \in \mathbb{R}^d$,

$$\left\|\nabla f(\mathbf{x}) - \nabla^2 f(\mathbf{x}^*)(\mathbf{x} - \mathbf{x}^*)\right\| \leq C_m \|\mathbf{x} - \mathbf{x}^*\|^2.$$

This inequality is similar to the assumption from Dieuleveut, Patel (2019) of uniformly bounded third derivatives. In the following theorem, we relax these assumptions and show that OSA achieves linear speed-up under considerably milder assumptions.

Before proceeding, let us define the step-size sequence $\{\theta_t\}$ as

$$\theta_t = \begin{cases} \frac{1}{L}, & \text{for } t = 0, \ldots, t_0 - 1, \\ \frac{2t}{\mu(t+1)^2}, & \text{for } t \geq t_0, \end{cases} \tag{2}$$

where $t_0 = \lfloor 2L/\mu \rfloor$. Notice that $\theta_t \leq 1/L$ for all $t$.

**Theorem 2.** *Under Assumptions 1 (smoothness), 3 (PL condition), 4 (twice differentiability at the optimum) and 6 (sub-Gaussian noise) and with step-size sequence $\{\eta_t\} = \{\theta_t\}$ defined in (2), we have for $T \geq t_0$,*

$$\mathbb{E}\left[\left\|\bar{\mathbf{x}}^T - \mathbf{x}^*\right\|^2\right] \leq \frac{4\sigma^2}{3\mu^2 NT} + o\left(\frac{1}{T}\right).$$

We are thus able to relax the conditions from the earlier literature, which required everywhere or almost everywhere higher derivatives with uniform bounds on third derivatives to merely twice differentiability at a single point. As a bonus, we also replace strong convexity with the PL condition.

This theorem is proved in Appendix C. The main difference between Theorem 2 and Corollary 1 is that Theorem 2 shows a linear speed-up with only one communication round but with slightly more restrictive assumptions such as sub-Gaussian noise model and twice-differentiable objective function at the optimal point. On the other hand, our results for OSA only require the PL-condition instead of strong convexity.

## 4 Numerical experiments

To verify our findings and compare different communication strategies in Local SGD, we performed the following numerical experiments, using an Nvidia GTX-1060 GPU and Intel Core i7-7700k processor.

### 4.1 Quadratic function with strong-growth condition

As discussed in Zhang et al. (2016); Dieuleveut, Patel (2019), under uniformly bounded variance, one-shot averaging performs asymptotically as well as mini-batch SGD, at least for quadratic functions.

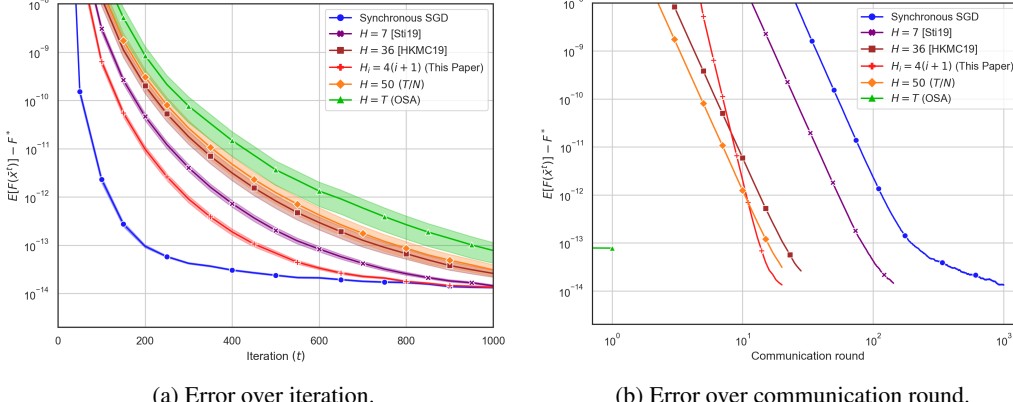

(a) Error over iteration.  (b) Error over communication round.

Figure 1: Minimizing (3) using Local SGD with different communication strategies. Figures (a) and (b) show the error over iteration and communication rounds, respectively.

Therefore, to fully capture the importance of the choice of communication times $\mathcal{I}$, we design a *hard* problem, where noise variance is uniform with strong-growth condition, defined in Assumption 5. Let us define,

$$F(\mathbf{x}) = \mathbb{E}_\zeta f(\mathbf{x}, \zeta), \qquad f(\mathbf{x}, \zeta) := \sum_{i=1}^{d} \frac{i}{2} x_i^2 (1 + z_{1,i}) + \mathbf{x}^\top \mathbf{z}_2, \qquad (3)$$

where $\zeta = (\mathbf{z}_1, \mathbf{z}_2)$ and $\mathbf{z}_1, \mathbf{z}_2 \in \mathbb{R}^d$, $z_{1,i} \sim \mathcal{N}(0, c_1)$ and $z_{2,i} \sim \mathcal{N}(0, c_2)$, $\forall i \in [d]$, are random variables with normal distributions. We assume at each iteration $t$, each worker $i$ samples a $\zeta_i^t$ and uses $\nabla f(\mathbf{x}, \zeta_i^t)$ as a stochastic estimate of $\nabla F(\mathbf{x})$. It is easy to verify that $F(\mathbf{x})$ is 1-strongly convex and $d$-smooth, $F^* = 0$ and $\mathbb{E}_\zeta[\|\nabla f(\mathbf{x}, \zeta) - \nabla F(\mathbf{x})\|^2] = c\|\nabla F(\mathbf{x})\|^2 + \sigma^2$, where $c = c_1$ and $\sigma^2 = dc_2$.

We use Local SGD to minimize $F(\mathbf{x})$ using different communication strategies, namely, synchronized SGD where $H = 1$, $H \approx \sqrt{TN}$ Stich (2019), $H \approx (TN)^{1/3}$ Haddadpour et al. (2019), $R = N$ with constant $H \approx T/N$ Stich, Karimireddy (2019); Khaled et al. (2020) and finally the communication strategy proposed in this work with $R = N$ and linearly growing $H_i$ local steps. We used $N = 20$ workers, $T = 1000$ iterations, $c_1 = 1.0$ and $c_2 = 10^{-10}$ with $d = 3$ and step-size sequence $\eta_t = 3/(\mu(t+1))$. To estimate the expected value of errors, we repeated the optimization using each strategy 100 times and reported the average and 1-standard-deviation error bar in Figure 1.

We make the following observations from Figure 1:

- Figure 1(a) shows that a communication strategy with increasing local steps (proposed in this work), outperforms all the other methods, both in transient and final error performance, specifically the one with the same number of communication rounds evenly spread throughout the whole optimization. This confirms the advantage of more frequent communication at the beginning of the optimization, especially when the ratio of $c$ to $\sigma^2$ in the noise with growth condition is large (see the definition in Assumption 5).

- Figure1(b) shows that our communication method uses fewer communication rounds, 20 versus 28 (Haddadpour et al., 2019), 143 (Stich, 2019) and 1000 rounds for synchronized SGD.

- OSA appears to perform relatively well despite using only one communication round, though not quite as well as other methods. This shows that the choice of communication is important in this experiment. In other words, it is not true that the success of our communication strategy is merely a byproduct of the experiment design, where any communication strategy, as long as it communicates at least once, will succeed.

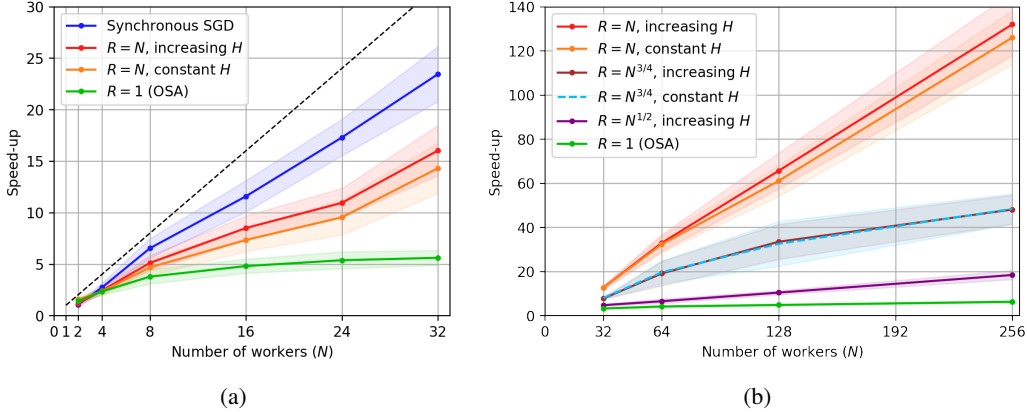

(a)                                                    (b)

Figure 2: Speed-up curves for different communication strategies, over different ranges of $N$ and $T$. Figure (a) establishes the linear speed-up of local SGD with $R = N$ communication rounds as well as failure of OSA to achieve speed-up even with small number of workers $N \leq 32$ over $T = 1000$ iterations. Figure (b) additionally plots speed-up curves for $R \approx N^{3/4}$ and $R \approx N^{1/2}$ for larger values of $32 \leq N \leq 256$ and $T = 8000$.

## 4.2   Speed-up curves

In this experiment, we minimize a one-dimensional function defined as,

$$F(x) = \begin{cases} \frac{1}{2}x^2, & x \leq 0, \\ x^2, & x > 0, \end{cases} \tag{4}$$

using Local SGD with gradients corrupted by a normal noise $\mathcal{N}(0, \sigma^2)$. We chose this specific cost function since it is not twice continuously differentiable at the minimizer $x^* = 0$ and does not satisfy Assumption 4 required by Theorem 2 for OSA to achieve linear speed-up. The results of this experiment will help us understand whether twice differentiability is a necessary assumption for OSA to obtain a linear speed-up.

The speed-up curve is derived by dividing the *expected* error of a single worker SGD by the *expected* error of each method at the final iterate $T$, over different number of workers $N$. Thus in the case where the error decreases linearly in the number of workers, we should expect to see a straight line on the graph.

We plot the speed-up curve for $N$ workers using different communication strategies: synchronized SGD, $R = N$ communication rounds with linearly increasing number of local steps $H_i$, $R = N$ with constant number of local steps $H \approx T/R$, as well as OSA with only $R = 1$ communication at the end. We use the step-size sequence $\eta_t = \min\{1/L, 2/(\mu(t+1))\}$ with $\mu = 1, L = 2$, and $\sigma = 8$, $T = 1000$.

Our results in Figure 2(a) show that Local SGD with $R = N$ (increasing or constant $H$) achieves linear speed-up in the number of workers, albeit with a worse constant compared to synchronized SGD. However, OSA fails to scale as $N$ increases. This suggests that the condition of twice differentiability (Assumption 4) is necessary for Theorem 2, as this function satisfies all the other assumptions of that theorem.

While our theoretical results provide only an upper bound on $R$ to achieve linear speed-up, this setting gives us a chance to find out if smaller number of communication rounds are enough. Therefore we repeat this experiment for larger number of workers $N$ and $T = 8000$, using $R \approx N^{3/4}$ and $R \approx N^{1/2}$ communication rounds. Our results in Figure 2(b) show that $R = N$ clearly achieves speed-up for larger values of $N$, as expected and $R = 1$ and $R \approx N^{1/2}$ fail to speed-up. However, $R \approx N^{3/4}$ also struggles to *linearly* speed-up in the number of workers, as the slope of the speed-up curve declines with $N$ increasing. It would be of interest to look into a more granular choice of communication rounds such as $R \approx N^{0.9}$ or even $R \approx N^{0.99}$ but this would require much larger values of $N$ and $T$ and thus more repeated simulations, which is beyond our computational resources, which were already exhausted by generating Figure 2(b).

It is worth mentioning that in both experiments of Figure 2(a) and 2(b), $R = N$ with increasing $H$ outperforms the one with constant $H$, even though the noise model used in this experiment is simply uniformly bounded, without strong-growth condition. This further endorses the use of more frequent averaging at the beginning of optimization, when paired with decreasing step-size sequence.

### 4.3 Regularized logistic regression

We also performed additional numerical experiments with regularized logistic regression using two large real datasets: (i) a national dataset (NSQIP) of surgeries performed in the U.S., seeking to predict short-term hospital re-admissions, which consists of **722101** data points (surgeries) each characterized by $d = 231$ features, (ii) the a9a dataset from LIBSVM (Chang, Lin, 2011) which includes **32561** data points with $d = 124$ features. The results of these experiments are presented and discussed in Appendix A.

## 5  Conclusion

In this work, we studied the communication complexity of Local SGD and provided an analysis that shows that $R = \Omega(N)$ number of communication rounds, independent of the total number of iterations $T$, is sufficient to achieve linear speed-up. Moreover, we showed only a single round of averaging is needed provided that the objective is twice differentiable at the optimum point. This assumption appears to be necessary, as our simulations show that not only one-shot averaging but using $N^{1/2}$ or $N^{3/4}$ communications in local SGD fails to deliver linear speed-up on a simple example which is not twice differentiable at the optimum.

## Acknowledgments and Disclosure of Funding

The research was partially supported by the NSF under grants DMS-1664644, CNS-1645681, ECCS-1933027 and IIS-1914792, by the ONR under grants N00014-19-1-2571 and N00014-21-1-2844, by the ARO under grant W911NF-1-1-0072, by the NIH under grants R01 GM135930 and UL54 TR004130, by the DOE under grants DE-AR-0001282 and NETL-EE0009696, and by the Boston University Kilachand Fund for Integrated Life Science and Engineering.

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
