# OpenReview forum: "Communication-efficient SGD: From Local SGD to One-Shot Averaging"
_NeurIPS.cc/2021/Conference — NeurIPS 2021 Poster_

### Official Review · Reviewer_d1us · 2021-07-15

**Rating:** 7
**Confidence:** 4

**Summary:**

This paper investigates the gap between local SGD and One-Shot averaging in communication efficacy and linear speed-up with respect to the number of workers. Through extensive analysis, they show that they can tighten the bound for the number of communication required for local SGD to $\Omega\left(N\right)$, which is the number of workers. They have proved this bound for strongly convex and objectives with PL condition, where the strong convexity can be a special case of it. They use the idea of adaptive local step to have a more frequent communication at the beginning of the training and less frequent one at the end. In their experimental section, they show that their proposal can converge to the same rate as other local SGD approaches and Sync SGD using  $\Omega\left(N\right)$ communication rounds.

**Limitations And Societal Impact:**

I do not think that this will be applied to this paper.

**Main Review:**

This paper bridges the gap from the local SGD and one-shot averaging to tighten the communication bound required for linear speed-up in local SGD. They are using heuristic ideas from Haddadpour et al. (2019) to use an adaptive and increasing number of iterations for each round of communication. Their theoretical analysis proves the effectiveness of this approach for strongly convex and PL-condition objectives. From the theoretical perspective, the contribution of this paper is important for tightening the bounds on the number of communication rounds required for linear speed-up, and hence it is a significant improvement over the previous works in this domain. The claims of the paper are clear and the comparison with other SOTA methods in this domain is fair. The empirical results are consistent with the theoretical results claimed in the paper and show the effectiveness of the proposed approach. I have some questions regarding the claims in the paper:

1. Although the theoretical results are focused on strongly convex and PL-condition objectives. However, I would like to know how this approach would work in non-convex objectives using local SGD.
 2. The theoretical results are mainly based on the adaptive local step sizes, however, in practice the difference between adaptive and non-adaptive cases are marginal. Could you please explain the difference and the significance of the adaptive local step?
3. How does the value of $a$ is decided? Are there any theoretical bounds on the size of local steps for this algorithm?


**Time Spent Reviewing:**

4

---

> ### Author Response · Authors · 2021-08-10
> **Response to reviewer d1us**
>
> We want to thank the reviewer for the positive and balanced assessment of our work. The questions asked by the reviewers are things we have been considering in the context of future words. Our answers are below:
>
> (1) We believe in general our techniques should be applicable to the non-convex case, though this requires a completely new analysis with almost zero carryovers from the technical content of the present paper.
>
> The high-level intuition is that communication early is better than communication later because of the decaying step-size: earlier, the nodes are farther apart rather than later, so that it is more important to bring the nodes closer together right away. This has been ignored by all the works which consider fixed-length communication intervals. We believe one should be able to improve most local SGD strategies by following this idea.
>
> The improvement in the strongly convex case was O ( poly log T) because the complexity was already known to be O( poly log T) from previous work, and reduced to O(1) in this paper. In the non-convex case, an O( poly log T) communication complexity is not known, so the improvement is likely to be different.
>
> More concretely, one of the authors of this paper has another manuscript in submission that shows that with non-convex function and **heterogeneous data**, $T^{1/4}$ communications suffice. Specifically, in the non-convex case with noise in gradient evaluation, one can only achieve $\sqrt{N}$ speedup in the number of workers rather than linear speedup in the strongly convex case, and this is indeed achieved by $T^{1/4}$ communications. In the non-convex and **homogeneous** case, a similar result was obtained by (Gauri & Joshi 2018). We are currently looking into the speedup that can be obtained with increasing communication intervals, but, as mentioned above, this requires a completely new analysis.
>
> (2): We assume that by local steps the reviewer here referred to the lengths of the communication intervals (because the step-sizes themselves are not adaptive). We actually want to disagree with the reviewer a little: in figure 1a, the benefit of having increasing intervals vs fixed is a multiplicative factor of ~2.5 in performance (red curve vs orange curve). This seems to us like a quite good gain from the simple tweak of letting the size of your communication intervals grow (while keeping the number of communications the same)!
>
> (3) There is a formula for the parameter a on line 215 of the paper (it is easy to miss). Examining that formula, one sees that the sizes of the intervals grow from $\approx 2T/R^2$ initially to $\approx 2T/R$ from the last one.

---

> > ### Comment · Reviewer_d1us · 2021-08-29
> > **Maintain my score**
> >
> > Thanks to the authors for their response. I think they have addressed my concerns. It seems that as other reviewers pointed out as well, this paper presents a valid theoretical understanding of the OSA problem, which is a valuable first step in this direction. As I have personally experienced the OSA problem in practice, I attest that in the cases this paper has worked on using adaptive step size or adaptive batch size is very useful for convergence of OSA. In the case of a non-convex setting or heterogeneous local SGD, this might not be true, and there might be a residual error at the end, which requires more rounds of communication. Hence, I believe providing a nice theoretical background for this problem is in the right direction. Although as mentioned by other reviewers, the sub-gaussian noise assumption or poly-log T improvement might be marginal improvements, I think this paper still is worthy of publication and will be beneficial to the community working in this direction.

---

### Official Review · Reviewer_ReEQ · 2021-07-16

**Rating:** 5
**Confidence:** 2

**Summary:**

In this paper, the authors study the theoretical speedup of local SGD and one-shot averaging. A synchronization scheme that communicates more frequently at the beginning is also suggested. Some simple numerical experiments are provided to verify the theoretical results.

**Limitations And Societal Impact:**

1. This paper uses the homogeneous data assumption. However, most of the recent deep-learning applications require non-iid settings, especially for large-scale problems where communication reduction via local SGD is more useful. Since the results of this paper only applies to the iid cases, such results are less useful in practical work.
2. Some strong assumptions such as strong convexity or PL condition are used. It will be more interesting to have the theoretical results under the assumption of smoothness without strong convexity or PL.

**Main Review:**

The paper is well-written. I recommend highlighting the communication scheme in Theorem 1 by putting it into a standalone definition, which will make it more friendly to the readers.
This paper focuses on the theoretical analysis of the speedup of local SGD and one-shot averaging, and provides some improved results compared to the previous work.
Some typos: line 140: sever -> server


**Time Spent Reviewing:**

4

---

> ### Author Response · Authors · 2021-08-09
> **Response to Reviewer ReEQ**
>
> We want to thank the reviewer for their positive comments. Regarding the two criticisms, our response is below:
>
> (i) We believe in the long run the homogeneous case will be the more important one, even acknowledging the correctness of what the reviewer writes about recent applications of deep learning. Let us explain why.
>
> The homogeneous case makes sense when you can give the same data set to all the nodes. The heterogeneous case makes sense when you cannot. Now consider the typical scenario for training a machine learning method: you want to parallelize computation among nodes in a cluster (e.g., in contrast to the federated learning scenario where your nodes are in the cloud and may have privacy concerns).
>
> Why can't you give the data set to all the nodes? The problem is storage.  One wants to use graphics cards that have storage of several gigabytes - but the best datasets are far larger. For example, Imagenet is ~150 GB.
>
> However, the storage on GPUs is increasing rapidly. We are writing this, for example, on a recently purchased computer with an RTX 3090, which is a GPU with 24 GB of memory. In general, there are now many GPUs with this much memory (e.g.,  https://pangoly.com/en/hardware/vga/gpu-memory-size/24gb ). The next generation of GPUs seems to have 48 GB, see e.g., https://www.tomshardware.com/news/nvidia-rtx-a6000-48gb-benchmarked
>
> On the other hand, data set size is not increasing nearly as fast -- for example, Imagenet has been the standard benchmark for a while now. It seems likely that we are a couple of doublings away from the point where even the largest data sets will fit on a GPU (and this is not even talking about smaller data sets such as CIFAR, which are <200 MB, and which have been able to fit entirely on GPUs for a long time now).
>
> So whether the homogeneous or heterogeneous assumption makes more sense depends on whether you can fit the entire data set on a single GPU and the prevailing trends suggest that this will very soon be the case for many of even the largest data sets. Because **the results are much better** in the homogeneous case, we believe the homogeneous case is worthy of special study -- so that, we would submit to the reviewer that our focus on the homogeneous case is not a weakness.
>
> (2) The strongly convex case is particularly interesting since the original Local SGD paper [Stich, ICLR 2018] introduced Local SGD in the strongly convex case. Additionally, the particular application we consider in this paper -- logistic regression for predicting hospitalizations -- is strongly convex (see Eq. (5) in the supplementary information) -- and for hospitalization prediction, logistic regression rather than neural networks is state of the art. See our related discussion in the response to Reviewer 7aRM. In general, if images or text are not to be processed and one deals with tabular data (Electronic Health Records with continuous and categorical variables as in the case of hospitalization prediction) neural nets (MLP) often perform worse than logistic regression, SVM, or other (linear) methods with careful variable selection.

---

> > ### Comment · Reviewer_ReEQ · 2021-08-31
> > **response to authors' feedback**
> >
> > Thanks for the feedback.
> > 1. For the non-iid settings, the small datasets like CIFAR usually doesn't require distributed training. As the authors mentioned, ImageNet is used as a benchmark for a while, thus cannot represent how the dataset sizes increase these days. In fact, the datasets used in BERT or GPT-3 are extremely large, and these are just for academia. In industry the datasets could be even larger. That's one of the reason why non-iid settings are important. For the theoretical analysis, theories in non-iid settings could usually cover or be easily converted to the ones in iid settings. Thus, the theoretical analysis in iid settings is relatively weak.
> > 2. Right after the publication of the original Local SGD paper [Stich, ICLR 2018], there is a paper [Yu et al. (2019)] for non-convex and non-iid settings.
> > In overall, the authors' feedback is not very convincing to me and I still think the contribution is limited. Thus, I'm going to keep the current rating.

---

> > > ### Author Response · Authors · 2021-09-01
> > > **Response**
> > >
> > > Thank you for your reply. A couple of counterpoints are below:
> > >
> > > > For the theoretical analysis, theories in non-iid settings could usually cover or be easily converted to the ones in iid settings.
> > >
> > > On the contrary: in the non-i.i.d. case, our results would impossible. We know this from e.g., the lower bounds proved by Arjevani and Shamir in https://arxiv.org/pdf/1506.01900.pdf
> > >
> > > Specifically, Arjevani and Shamir prove a lower bound showing that, if the functions are non-i.i.d. and arbitrarily unrelated, then the number of communications to compute an epsilon-accurate solution will grow like log(1/epsilon). This lower bound holds for any algorithm based on taking linear combinations of gradients, Hessians, inverses of Hessians, optimization problems arising from all these, i.e., almost any optimization method you can think of. Further, this lower bound holds even if there is no noise in the gradient evaluations.
> > >
> > > Now in our setup gradients are corrupted by noise, so our problem is harder in that sense; but we are able to find an epsilon-accurate solution in a number of iterations that **does not depend at all on epsilon**. The difference is in the  i.i.d. assumption. **Thus, we see that theoretical results in the convex + non-i.i.d. setting cannot be converted to obtain our main results here.** The same also holds for Theorem 2 on one-shot averaging, which fails in the non-i.i.d. setting.
> > >
> > > > Right after the publication of the original Local SGD paper [Stich, ICLR 2018], there is a paper [Yu et al. (2019)] for non-convex and non-iid settings.
> > >
> > > We cite that work; it is a beautiful paper, but on our reading it requires T^{3/4} communications for T iterations (the discussion on the right-hand side of page 5 in that paper suggests taking the inter-communication interval to be T^{1/4}). Compare that with our result, which has no dependence on T at all in the number of communications.
> > >
> > > In general, stronger assumptions should lead to better results, and this is something that should be explored by the ML community. We do not see why the existence of results under non-convexity and non-i.i.d. sampling, should invalidate work obtaining better results under additional assumptions.

---

> > > > ### Author Response · Authors · 2021-09-01
> > > > **Typo**
> > > >
> > > > > number of iterations that does not depend at all on epsilon
> > > >
> > > > Should be: "number of _communications_ that does not depend at all on epsilon."

---

### Official Review · Reviewer_7aRm · 2021-07-17

**Rating:** 6
**Confidence:** 4

**Summary:**

## Update after rebuttall

I am satisfied with the authors' response and elevate my score to 6.

##

This paper presents several new theoretical results on the convergence of Local SGD as well as numerical experiments to check the tightness of the theory. In particular, the authors try and remove logarithmic terms from the bounds of Local SGD by making the communication less frequent as the iteration counter increases. The main results of the paper are:

1. Theorem 1 gives convergence for strongly convex functions under uniform-with-strong-growth noise. I argue in my detailed review that this result looks suboptimal.

2. Theorem 2 gives a result for PL functions with sub-Gaussian noise for one-shot averaging. This result is interesting as it promises a linear speed-up, although under somewhat restrictive assumptions.

3. The experiments show that linear speed-up is lost when the number of communication rounds is chosen smaller than predicted by the theory of Theorem 1.

I tend to vote for rejection of this work, even though I appreciate some of the results obtained here. My main concerns are 1) applicability of the message to neural network training, which is highly relevant to FL; 2) the theoretical improvements appear to be small; 3) some small flaws in the proofs.

**Limitations And Societal Impact:**

yes

**Main Review:**

## Weaknesses
1. From the numerical studies on Local SGD in training neural networks one can see that the proposed ideas are unrealistic. For instance, by reading the paper (Lin et al. "Don't Use Large Mini-Batches, Use Local SGD") one can mention that a relatively small number of local steps, such as 16, is used in practice. Increasing this number linearly as in Theorem 1 seems exotic, and without a numerical study by the authors I cannot conclude positively on the promise of such ideas.

2. Theorem 1 does not look very good to me. Since beta is lower bounded by kappa^2, it also holds that the stepsize is smaller than 1/(kappa * L), which does not seem to be a good assumption. Moreover, the upper bound in Theorem 1 includes the term kappa^2/(R*T), which seems to be kappa times worse than the similar term kappa*H/T^2 in Corollary 1 of Khaled et al. I am also puzzled as to why the authors used beta=1 in their experiments as explained in Appendix A, which is clearly against their theoretical results.

3. The proofs are a bit unexplained, some definitions seem missing. For instance, I guess from some equations that g_i^t (without any hats or bars) is the expectation of \hat g_i^t, but I didn't find the definition in the text. As far as I can see, the epsilon in the proof of Lemma 4 was not introduced either.

I did not understand the first step in equation (16). Why does it hold that sum_i ||g_i^t - overline g^t||^2 <= sum_i ||g_i^t - nabla f(\overline x^t)||^2 ?

## Proposed improvements
1. In the strongly convex case, the analysis could be done with the variance of gradients measured at the optimum instead of the global bound.
2. (Completely optional) I have a wild guess that additional results can be obtained under the setting of statistical similarity (Hendrikx et al., 2020), because for homogeneous data (sigma=0) the bounds significantly improve. Since the authors already perform analysis based on the Hessian at the optimum. I only suggest this in case the authors decide to resubmit their paper to a different venue or to write a new paper,  I do not ask the authors to do this during the rebuttal period.

Hendrikx et al., 2020, "Statistically Preconditioned Accelerated Gradient Method for Distributed Optimization"

## Missing references
"Parallel Gradient Distribution in Unconstrained Optimization" by O. Mangasarian (1995) is probably the first work to introduce the idea if Local SGD and do (asymptotic) analysis. I believe this paper should be cited

## Typos and minor comments
1. Abstract: I suggest writing Omega(poly(log T)) instead of Omega(polynomial in log T)
2. Abstract line 11: "state-of-the-art" is adjective and "state of the art" is a noun, so please fix it
3. Page 2, line 64: "strongly-convex" -> "strongly convex"
4. Page 3, line 89: "at the optimizer" -> "at the optimum"
5. "Charles, Papailiopoulos (2018) shows" (and many other examples): it feels rather strange that you refer to papers with multiple authors as to a singular noun.
6. "Lohasiewicz" -> "{\L}ojasiewicz"
7. Page 4, line 208: "for t\in I and i." What does this mean?
8. Page 8, line 293: "to obtain a linear up" -> "to obtain a linear speed-up"
9. Page 15, line 552: "the the"
10. Page 17, equation (13) and the lines after: "f(x^t) - F*" -> "f(x^t) - f*", "smoothness of F" -> "smoothness of f"

**Time Spent Reviewing:**

6

---

> ### Author Response · Authors · 2021-08-09
> **Response to Reviewer 7aRM**
>
> We want to thank the reviewer for the detailed and thoughtful review. We will certainly incorporate all the comments here into the manuscript. Regardless of the outcome of this submission, we appreciate the careful reading our paper has received.
>
> Where we disagree fundamentally with the reviewer is on point 1. It is true that neural networks are important -- but why should that be a primary criterion for the evaluation of our work? All of our examples are on logistic regression. And while we all know there are spectacular success stories for neural networks, there are many ML applications where logistic regression is the state-of-the-art. Particularly in health-related application settings where interpretability is important, logistic regression with careful variable selection (e.g., statistical, recursive, etc.) is much more preferable to a neural-network black-box. A recent case in point: A large number of COVID-19 calculators have been published (e.g., to predict hospitalizations, ICU admission, use of a ventilator, etc.) and the majority used logistic regression. Another interesting case in point: the 10-year risk for cardiovascular disease automatically computed for everyone after a cholesterol check has been computed with logistic regression (by the Framingham Heart Study).
>
> Moving on, the reviewer's point 2 is valid. While we show that our communication bounds are conceptually better in terms of no scaling with T whatsoever, this comes at a cost: some of the transient terms are worse compared with the previous literature. But we would point out that it may not even be possible to improve every aspect of the bounds, and so this seems like something better left for future work. Indeed, our paper answers what seems to us to be a meaningful question -- does the number of communications need to depend on T at all? -- which seems like a worthwhile contribution even if some of the transient terms have worse scaling with condition number.
>
> Finally, on point 3, the reviewer correctly guessed the definition of g_i-hat, defined on line 567.  Regarding the epsilon in the proof of Lemma 4: did the reviewer get the letter or the lemma number wrong? We don't see an epsilon in the proof of Lemma 4.
>
> > I did not understand the first step in equation (16). Why does it hold that $\sum_i ||g_i^t - \overline g^t||^2 \leq \sum_i ||g_i^t - \nabla f(\overline x^t)||^2$ ?
>
> Let us first consider the case where all these variables are scalars. Then this is just a consequence of the general fact that
> $\sum_i (x_i - y)^2$
> is minimized when y is the average of the numbers $x_i$. The LHS is just this equation where we plug in y to the average; and on the right-hand side, we plug in y to be some other number.
> To obtain this fact for vectors, one can simply apply it component-by-component.

---

> > ### Comment · Reviewer_7aRm · 2021-08-10
> > **Thank you for the clarifications**
> >
> > The authors are right that the research is not all about neural networks. If the authors want to simply say that we should ignore them as an application and instead focus on models like logistic regression, I'm fine with that. I do want to note that based on the paper (Kairouz et al., "Advances and Open Problems in Federated Learning") it seems that logistic regression is of less interest. If we search for ``neural network``, we get 34 results, while ``logistic regression`` gives 2. However, my initial goal was to point out that the results are not useful for deep learning, not to say that there are no other applications.
> >
> > Thank you for acknowledging the shortcomings of your results with respect to kappa. I get your point that you try to understand a particular aspect of complexity. I am not so sure, though, that it is a good strategy to optimize one part of complexity in this work and leave improvements in other constants for future papers.
> >
> > Thank you for clarifying the derivation, this step turned out to be quite easy. I was confused because the inequality did not seem to use in any way the fact that on the right-hand side we have the gradient, but your argument actually holds for any vector, so now it makes perfect sense to me.

---

> > > ### Author Response · Authors · 2021-08-11
> > > **Brief response**
> > >
> > > Thanks for the reply; it seems that we largely agree at this point. We want to address a couple remaining points of disagreement:
> > >
> > > 1) We're not sure if what we are doing is really Federated Learning (FL),  since we are just looking at speedups that can be done by giving the same data set to many processors in the same cluster. Our understanding is that FL deals with having many geographically separated processors such as mobile phones which may be unreliable, delayed, have their own data, and have privacy concerns.  The point is that even if what we are doing is a part of FL, it is clear that FL is a much broader field, so that gauging the importance of an application in our setup by looking even at a canonical survey on FL may not be the right thing to do. That is, if X is a small subfield of Y, then looking at typical applications of Y will not give the right answer about typical applications of X.
> > >
> > > 2) The point about optimizing only one part of the system is fair. But consider that we are concerned with the dominant term as T goes to infinity, which is surely well-justified. After all, the analysis of asymptotics of dominant terms is quite common throughout optimization, statistics, etc.

---

> > > > ### Comment · Reviewer_7aRm · 2021-08-11
> > > > **We seem to agree**
> > > >
> > > > 1. I get what you are saying about FL. The reason I talked about FL is that I tried to think of a typical example where Local SGD could be useful. If we talk about centralized optimization of a logistic regression objective, maybe we should rather use some sort of asynchronous SAGA or accelerated SVRG algorithm. In other words, FL might not be the best setting for your method, but it's not clear what setting is the best otherwise.
> > > > 2. I think we are on the same page about this.

---

### Official Review · Reviewer_3FSZ · 2021-07-21

**Rating:** 6
**Confidence:** 5

**Summary:**

This paper analyzes local SGD with progressively decreasing communication frequency under the homogeneous data assumption. Unfortunately, the obtained convergence rates are worse than the best-known convergence rates for FedAC (Yuan and Ma) under the same set of assumptions. The paper also analyses One-shot averaging under a slightly different set of assumptions, showing it retains its linear convergence in settings less restrictive than pre-existing literature. Since the major result of this paper is worse than FedAC's guarantee, it is very incremental.

**References**

Yuan, Honglin, and Tengyu Ma. "Federated Accelerated Stochastic Gradient Descent." Advances in Neural Information Processing Systems 33 (2020).

**Limitations And Societal Impact:**

The authors have not compared to the most important baseline FedAC, which is detrimental for this paper.

**Main Review:**

It seems the authors are unaware of FedAC, which requires $N^{1/3}$ synchronization for a linear speedup, and is a first-order algorithm analyzed under strong convexity, smoothness and uniform bounded variance. This is my major concern with the paper, since proving a lower communication complexity for Local SGD was its main (claimed) contribution.

One-shot averaging is known to be minimax optimal for quadratic functions (after adding acceleration) (Woodworth et. al.). Previous works require higher-order assumptions (satisfied by quadratics) to show such speed-up for one-shot SGD. Thus, I understand the appeal to somehow push the result for OSA to a larger class of functions. But again, in absence of comparison to results from FedAC for both their A1 and A2 assumptions, I am hesitant to make a judgment on these results.

Finally, I don't think the real-data experiments demonstrate any statistically significant advantage. Moreover, without comparison to FedAC, I am not ready to buy any experimental claims here.

# After a discussion about FedAC

After the authors acknowledged that they missed a fair comparison to FedAC, and have listed some of its theoretical limitations, I am increasing my score to 5. If my concerns regarding empirical comparison to FedAC are resolved then I will further increase my score.

# Final Verdict

After the authors made the required empirical comparison with FedAC, I am more satisfied with the paper. I would urge the authors to include more experiments in the final version of the paper with a larger tuning grid, larger condition number, with clear comparison against FedAC and the other baselines. In light of all the back and forth discussion, as well as the proposed changes I am increasing my score to a 6 and am fine with the paper being accepted.

**References**

Woodworth, Blake, et al. "Is local SGD better than minibatch SGD?." International Conference on Machine Learning. PMLR, 2020.

**Time Spent Reviewing:**

2

---

> ### Author Response · Authors · 2021-08-09
> **Response to Reviewer 3FSZ**
>
> First, thanks very much for this review. Indeed, we were unaware of FedAC and this is obviously an essential paper for us to compare with. Lacking that comparison, the reviewer's rating on our paper as submitted is justified.
>
> Having said that, we spent the past week looking carefully at FedAC and we believe that a careful comparison is, in many ways, favorable to us:
>
> -- FedAC does not address the main question we are concerned with: can we do a number of communication rounds that does not scale at all with T? We do not want to communicate log T to some power number of times; we want to communicate a constant number times that does not depend on T.
>
> All the O's in FedAC are O-tildes that hide powers of log, so that this point is unaddressed there.
>
> -- The above is not a purely theoretical point. Here, for example, is a simulation for both algorithms in the FedAC paper vs. our method on our standard example: the discontinuous quadratic, see Eq. (4) in our paper. This is the most natural example for these methods, as it is the simplest example of a strongly convex function that is not twice differentiable (for twice differentiable functions, our paper shows just one communication is fine with OSA):
>
> https://anonymous.4open.science/r/Local-SGD-re-283F/figures/1d_quadratic_errors.png
>
> Here the number of workers is N=20. As can be seen, improving the scaling with the number of workers while adding logarithmic factors of T makes FedAC perform worse!
>
> -- FedAC, as written, appears to require knowledge of both the initial distance to the optimal point, and the optimality gap $F(w_0) - F^*$ of the function you are optimizing. This is unrealistic, as it requires you to know the optimal solution, which is the very thing you want to find,  ahead of time. Presumably, these could be replaced by upper bounds, but then all the convergence times derived in FedAC will scale with these upper bounds. This is not the case for our paper,
> which assumes no knowledge of these quantities or any upper bounds on them, and where the convergence time scales with $||w_0 - w^*||_2^2$ without assuming any knowledge of $w^*$.
>
> -- (Minor point) We assumed a much more realistic noise model, where the noise in querying the gradient at x is not a constant but depends on x. Under this so-called "strong growth condition" -- see our paper for definition -- the FedAC method with N^{⅓} communications does not appear to converge, here is our simulation:
>
> https://anonymous.4open.science/r/Local-SGD-re-283F/figures/iteration_skewed.png
>
> Here the number of workers is N=20, dimensions is d=3, and noise factors are c_1=1 and c_2=0.5 (see Section 4.1 of our paper for the settings of this experiment). Here we used the same exact step-size eta and parameters (gamma, beta, alpha) for FedAC-I and FedAC-II as defined in Equations 3.1 and 3.2 and Theorems B.1 and C.13 of the FedAC paper, respectively.
>
> -- When we try to simulate our method vs FedAC with the step-sizes recommended in the FedAC paper, we see that FedAC performs quite poorly:
>
> https://anonymous.4open.science/r/Local-SGD-re-283F/figures/speed-up.png
>
> https://anonymous.4open.science/r/Local-SGD-re-283F/figures/speed-up-lower.png
>
>
> We see that over the range of 2 workers to 256 workers, FedAC seems like almost the very worst thing you can do -- it does not seem to deliver any speedup at all. This should be contrasted with Figure 2b in our paper, which clearly shows the linear speedup. Intriguingly, we see a strong benefit of adding increasing communication intervals into Fed-AC (see the last figure).
>
> Incidentally, this does not necessarily contradict the theoretical results in the FedAC, which, we computed over this range, and which can be as big as $10^{11}$ for this example, so that it could simply be that it takes a while for the asymptotics to kick in. Those upper bounds are plotted here:
>
> https://anonymous.4open.science/r/Local-SGD-re-283F/figures/FedAC_potentials.png
>
> -- **MOST IMPORTANT POINT:** We believe there is a fundamental reason why FedAC seems to perform so poorly when we simulate it. It is the same reason why it has improved theoretical guarantees: it is an accelerated method. It is well known that accelerated methods can lack robustness in practice, and sometimes even in simple examples.  By contrast, our method is plain un-accelerated gradient descent. In general, it is well-accepted in optimization that analysis of un-accelerated methods remains of interest even if there are accelerated methods with improved theoretical guarantees -- since,  in practice, accelerated methods often underperform. We observed this with FedAC on the simplest possible example in this context, the quadratic with discontinuous second derivative (Eq. (4)).
>
> SUMMARY: we strongly agree with the reviewer that a comparison to FedAC, which we did not know about, is essential for this work. We have spent the last week looking into the FedAC paper and, as discussed above, the comparison is favorable to us in many aspects (even though  FedAC has a better bound in terms of scaling with the number of workers). Thus we would ask the reviewer to reconsider their rating in light of the above comparison.
>
>
> Other points:
>
> -- "Finally, I don't think the real-data experiments demonstrate any statistically significant advantage."
>
> Consider Figures 1a (left figure) and Figure 1b (right figure). We believe they show our method considerably outperforms all the others plotted there when you consider the figures together:
>
> (1) Obviously, synchronous SGD (which communicates at every step) will reduce the error faster than any other method (left figure). But then we compute the error per communication, it's pretty terrible (right figure).
>
> (2) Besides synchronous SGD, our method reduces the error fastest (left figure).
>
> (3) Our method reduces the error best on a per-communications basis (right figure).
>
> (4) Any method that is close to our method in the left figure is far from it on the right figure.
> For example, the purple curve is close to our method on the left figure, but far in the right figure.
>
> (5) Conversely, any method that is close to our method in the right figure is far from it in the left figure.  For example, the orange curve is close to our method in the right figure but not in the left figure (keep in mind the y-axis is log scale, so constant gains translate into bigger differences).
>
> Actually, the orange curve is the most natural comparison baseline, since it does as many communications as our method but with a fixed communication interval, and its final performance is approximately ~2.5x worse (left figure). This seems like a meaningful error reduction to get, considering that we only tweak the underlying algorithm slightly with our choice of inter-communication intervals.

---

> > ### Comment · Reviewer_3FSZ · 2021-08-16
> > **Comparison to FedAC**
> >
> > I thank the reviewers for promptly comparing to FedAC. I am looking at this version: https://arxiv.org/pdf/2006.08950.pdf
> >
> > **Experiments**
> >
> > > https://anonymous.4open.science/r/Local-SGD-re-283F/figures/1d_quadratic_errors.png
> >
> > This experiment is not convincing to me. If one changes the communication rounds between two algorithms it is difficult to compare them. And I understand that the motivation is FedAC's convergence guarantee, but it still doesn't make sense. Moreover, the comparison is on a synthetic data set with some additive noise on the gradients, which is also not a useful setting to consider. I encourage the authors to re-implement the setup in Fig. 3/Fig. 5 with the a9a dataset as  in the FedAC paper which equalizes the number of communication between different algorithms.
> >
> > > https://anonymous.4open.science/r/Local-SGD-re-283F/figures/iteration_skewed.png
> >
> > Again the same issue. FedAC-II communicates just 3 times. Please re-produce the above experiment, so that one can actually judge the relative performance across a choice of communication budgets for a fixed parallel run-time. But I do understand that your noise model is more general. It would be a good thing to have if your algorithm actually outperforms FedAC on a real data-set under their experimental setup.
> >
> > >https://anonymous.4open.science/r/Local-SGD-re-283F/figures/speed-up.png
> > https://anonymous.4open.science/r/Local-SGD-re-283F/figures/speed-up-lower.png
> >
> > Same issue here. FedAC communicates less than your algorithm. How then are you measuring the speedup? Is it the best suboptimality for a given amount of iterations? The ideal thing to do to measure this is Fig. 1 and 2 in the FedAC paper.
> >
> > > https://anonymous.4open.science/r/Local-SGD-re-283F/figures/FedAC_potentials.png
> >
> > Again not sure what's the setup here, communication rounds, local steps, number of machines?
> >
> > In short, I am not convinced by the experiments, and encourage the authors to replicate Fed-AC's experiments. Specifically, I am unhappy with (i) the synthetic data-sets, (ii) the really low and unequal number of communication rounds for FedAC, (iii) unclear setting in some figures, (iv) lack of important baselines such as large mini-batch SGD, accelerated minibatch SGD, accelerated local SGD as well as single machine SGD.
> >
> > All of the above are important baselines. See the optimal algorithm here: https://arxiv.org/abs/2102.01583
> >
> > The following would convince me: taking the experiments in FedAC and exactly reproducing them with your algorithm added. And if not for the current version, such a comparison would be necessary for any future version of the paper.
> >
> > **Other Issues**
> >
> > > FedAC, as written, appears to require knowledge of both the initial distance to the optimal point, and the optimality gap
> >
> > This is false, how does it require those? FedAC only requires one to tune the step-sizes and expresses other hyper-parameters in terms of that, c.f., eq. 3.1 and 3.2.
> >
> > > FedAC does not address the main question we are concerned with: can we do a number of communication rounds that does not scale at all with T? We do not want to communicate log T to some power number of times; we want to communicate a constant number times that does not depend on T.
> >
> > This is not a big issue. Consider T=10000 and N=1000, FedAC communicates $10(4)^p$, while you do 1000 times. Even in this setting, if FedAC's analysis has p= 3 power on its log factors, it seems better than your method?
> >
> > >  It is well known that accelerated methods can lack robustness in practice, and sometimes even in simple examples. By contrast, our method is plain un-accelerated gradient descent. In general, it is well-accepted in optimization that analysis of un-accelerated methods remains of interest even if there are accelerated methods with improved theoretical guarantees
> >
> > If it were the case that you beat FedAC in fair experiments (like I describe above) across a range of communication-computation budgets, then it would make sense to underline the limitation of accelerated methods. But currently, you don't do that. In fact, in real data experiments, there is no advantage of using your method as opposed to vanilla local SGD. And maybe even large mini-batch SGD (fewer iterations of MB-SGD with a larger batch size) which has the same communication and computation cost as local SGD. And I don't expect this algorithm to in fact beat large mini-batch SGD in all the settings.  Thus I am not ready to give the authors the benefit of doubt.
> >
> > > Consider Figures 1a (left figure) and Figure 1b (right figure).
> >
> > I was referring to a logistic regression data-set such as a9a which you experiment with, in the appendix. In a discussion with another reviewer, you mentioned how logistic regression for big data sets is very important. Then why have you not experimented in that setting, with some big data-sets like CovType?
> >
> > In addition, why was the Scaffold algorithm not compared to, which matches the guarantees of large mini-batch SGD? https://arxiv.org/pdf/1910.06378.pdf
> >
> > I do like the result in Theorem 2, it somewhat extends the result of Woodworth et. al. which you cite but don't discuss here. I would like to see the rate for accelerated OSA as well. We don't have a lower bound for the assumptions used here, so it would be good to develop as well.
> >
> > In light of all of this, I am not improving my score. This paper doesn't have an empirical benefit that is clearly exposed, or a theoretical one. And it lacks discussions on important first order lowers bound, and baseline algorithms. A significant re-writing is required.

---

> > > ### Author Response · Authors · 2021-08-19
> > > **Response**
> > >
> > > Please find a partial response below. We may add to this in the next few days to address any additional points:
> > >
> > > (1) We wrote:
> > >
> > > > FedAC, as written, appears to require knowledge of both the initial distance to the optimal point, and the optimality gap
> > >
> > > Reviewer replies:
> > >
> > > > This is false, how does it require those? FedAC only requires one to tune the step-sizes and expresses other hyper-parameters in terms of that, c.f., eq. 3.1 and 3.2.
> > >
> > > Our reply:
> > >
> > > On the contrary, this is true. The step-size in FedAC depends on the distance to the optimal solution, which is unknown. See Theorem B.1 where the step-sizes depend on eta, which in turn depends on Phi_0, which in turn depends on distance to the optimal solution and optimality gap. The same holds for Theorem C.1.
> > >
> > > The reviewer's response here is an evasion. It is a fact that the theorems in the FedAC paper require a step-size depending on the distance to the optimal solution; when we point this out, the reviewer replies that this is "False"  and Fed-AC just requires "step-size tuning." Sure...presumably the tuning effectively allows you to discover a bound on the distance to the optimal solution via trial and error or binary search.
> > >
> > > Furthermore, all claims of a linear speedup are SEVERELY weakened if one needs to tune the step-size like this because centralized gradient descent works with ~1/(mu*t) step-size does not need any additional tuning.
> > >
> > >
> > > (2) We write:
> > >
> > > > FedAC does not address the main question we are concerned with: can we do a number of communication rounds that does not scale at all with T?
> > >
> > > Reviewer replies:
> > >
> > > > This is not a big issue. Consider T=10000 and N=1000....
> > >
> > > We reply:
> > >
> > > Sure, you can game the comparison by picking a particular pair of N,T. You can get the opposite effect by picking an N,T pair that makes our method appear superior -- which will also be far more plausible from a practical point of view since using N=1,000 processors today is rare (e.g, far more plausible is to choose N=20 and N=100).
> > >
> > >
> > > (3)  Reviewer writes: I was referring to a logistic regression data-set such as a9a which you experiment with, in the appendix. In a discussion with another reviewer, you mentioned how logistic regression for big data sets is very important. Then why have you not experimented in that setting, with some big data-sets like CovType?
> > >
> > > As the reviewer points out, we have experimented with a big data set in the paper (where, due to smoothness, one-shot averaging performs the best, as predicted by our Theorem 2). We assume this refers to our response, where this experiment was not repeated with FedAC. Our reply is that this criticism is backwards.
> > >
> > > If you have a method which is supposed to perform well, the first example you should try it on is a quadratic. In this case, this should be something like f(x) = x^2 when x>0, (1/2) x^2 when x < 0, since, if the second derivative exists at the optimal solution, one-shot averaging will  already give the desired linear speedup as our Theorem 2 shows and  there's no need to look at any different method. As for the data points themselves, one can just generate them randomly.
> > >
> > > Now IF the method works on a simple quadratic, THEN it makes sense to try it out on real world data sets. If we show that a method does not seem to deliver gains on a quadratic (at least until N=256 nodes), it does not make sense to reply "why didn't you try it on a real world data set?"
> > >
> > >
> > > (4) Reviewer writes: "If one changes the communication rounds between two algorithms it is difficult to compare them. And I understand that the motivation is FedAC's convergence guarantee, but it still doesn't make sense. Moreover, the comparison is on a synthetic data set with some additive noise on the gradients, which is also not a useful setting to consider
> > >
> > > We reply:
> > >
> > > There's nothing difficult about it. One can plot, as we do, error per iteration and error per communication in two different graphs. This should give you a complete picture of how well the method performs on two different metrics.
> > >
> > > Again, the "synthetic data set" here is just a piecewise quadratic with randomly generated points, i.e., the simplest possible example you can write down which is strongly convex, with Lipschitz gradient, but not twice differentiable at the optimal solution.
> > >
> > > (5)  Reviewer writes: "I encourage the authors to re-implement the setup in Fig. 3/Fig. 5 with the a9a dataset as in the FedAC paper which equalizes the number of communication between different algorithms."
> > >
> > > Equalizing the number of communication is a plausible comparison, but it is not the only way the algorithms can be compared, and our method of comparison is just as good if not better.
> > >
> > > Contrary to what the reviewer writes, there's no difficulty comparing methods with different #communications. Indeed, consider one-shot averaging vs synchronous SGD on a smooth problem, shown in Figure 3b in our paper. From Theorem 2 we know they should be roughly the same, and indeed we see the performance per iteration is nearly identical...even though the former does 1 communication and the latter does N communications. We immediately conclude the former is better (similar error with fewer communications).
> > >
> > > Now let's turn to the comparison with FedAC. In the very first graph of our response, we showed that FedAC underperforms one-shot averaging on the quadratic mentioned above in BOTH error/iteration and error/communication. In other words, it does worse than OSA even though it does more communications. As can be seen, comparing here is not difficult at all.
> > >
> > >
> > > (4) Reviewer writes: "If it were the case that you beat FedAC in fair experiments (like I describe above) across a range of communication-computation budgets, then it would make sense to underline the limitation of accelerated methods. But currently, you don't do that. "
> > >
> > > We reply:
> > >
> > > Sure, but adding a discussion on un-accelerated vs accelerated methods is a very small change to make.
> > >
> > > (5) Reviewer writes: "Again the same issue. FedAC-II communicates just 3 times..."
> > >
> > > And one-shot averaging communicates just one time, which doesn't seem to prevent it from significantly beating FedAC on the example we gave. Also, FedAC communicates more on the graph which contains N=256.
> > >
> > >
> > >
> > > Summary:
> > >
> > > Our main takeaway from this discussion is that the reviewer is rather biased in favor of FedAC. The responses that a step-size which depends on the distance to the optimal solution does not actually depend on it but just requires "step size tuning" is particularly puzzling and troublesome.
> > >
> > > To be blunt, we have been looking at the FedAC paper and we think it is a brilliant piece of work; it will take us some time to absorb it, and we look forward to studying it further. However, like all research,  it is not without limitations, the main of which appears to be "step size tuning"/knowledge of the distance to the optimal solution issue. It also seems to perform poorly on the simplest possible example, which is intriguing. Its existence should not preclude research on un-accelerated methods which, besides their theoretical guarantees,  give the promised speedup on such an example, e.g., this work. We hope that all reviewers and the area editor will consider these very clear-cut issues we outline and will give our paper a fair chance.

---

> > > > ### Comment · Reviewer_3FSZ · 2021-08-19
> > > > **Response to authors 1/2**
> > > >
> > > > Thank you for your clarification.
> > > >
> > > > > On the contrary, this is true. The step-size in FedAC depends on the distance to the optimal solution, which is unknown. See Theorem B.1 where the step-sizes depend on eta, which in turn depends on Phi_0, which in turn depends on distance to the optimal solution and optimality gap. The same holds for Theorem C.1.
> > > >
> > > > Ok, I admit that I was wrong here, and apologize for the confusion. It was partially my mistake and partially how the authors in the FedAC paper chose to present and discuss their results! I didn't carefully look at the **full version** of the theorem in their appendix. To obtain the convergence rate they obtain, one needs to use a step-size which could depend on the problem-dependent parameters. In fact, it is slightly misleading to hide this dependence in log factors in the main version of their paper as they choose to do. Even if the dependence is only through logarithmic terms, and the learning rate has to be tuned regardless of what the theory says, this is a limitation of FedAC's analysis. Having said that, it is not unusual to assume a bound on the diameter of the parameter space in which case one shouldn't discount the result in FedAC too harshly.
> > > >
> > > > > The reviewer's response here is an evasion. It is a fact that the theorems in the FedAC paper require a step-size depending on the distance to the optimal solution; when we point this out, the reviewer replies that this is "False" and Fed-AC just requires "step-size tuning." Sure...presumably the tuning effectively allows you to discover a bound on the distance to the optimal solution via trial and error or binary search.
> > > >
> > > > I can assure you it was not an evasion, just a misunderstanding of their result. Moreover "This is false, how does it require those?" has a question mark, where I am qualifying my statement by specifying which equation I am looking at, and also citing the version I am looking at. The intention was to continue that discussion and not indulge in ad hominem! Having said that we are on the same page now, I do think this is a limitation of FedAC and a fair point. It should of course be discussed in a future version of your paper while comparing to FedAC. But I'd like to point out that in your theorem 2, your step size also depends on strong convexity and smoothness, which are also problem-dependent constants (albeit don't require the knowledge of $w^\star$). But I don't view that too harshly, because the step-size has to be tuned in practice, so assuming the existence of some optimum step-size is not as restrictive in my opinion. If it were the case that your method is more robust to the step-size selection, that would have been a different issue altogether. To clarify again, I was not putting away a dependence under the rug due to inevitable tuning, but we were referring to different equations before.
> > > >
> > > > > Sure, you can game the comparison by picking a particular pair of N,T. You can get the opposite effect by picking an N,T pair that makes our method appear superior -- which will also be far more plausible from a practical point of view since using N=1,000 processors today is rare (e.g, far more plausible is to choose N=20 and N=100).
> > > >
> > > > I was not trying to game the comparison, but only trying to underline that a logarithmic dependence on T, might be more desirable than an additional $N^{2/3}$ dependence, for a reasonable choice of the N, T. As reviewer 7aRm pointed out, while you might not want to target Federated Learning as a use case, it is unclear what else is a better use case. And FL is characterized by a large number of user devices, hence my example. In fact, it is not uncommon for the number of machines to run in thousands.
> > > >
> > > > I do understand your motivation to get a rate completely independent of $T$. But I need to judge the paper based on what was submitted, where you do highlight that having a $\Omega(N)$ rate independent of $T$ is a contribution. And in absence of FedAC everything looks fine, but introducing it into the picture makes the comparison more nuanced. It is important to have a discussion of what is more desirable: getting rid of polylogarithmic dependence on T or polynomial dependence on N, and under which settings. I incline more with the latter. I also hope that the authors understand that with FedAC in the picture, many areas of their writing would need to be changed/tone downed/or written with a disclaimer. And this is why it is difficult to judge the submitted version only with FedAC in the picture.
> > > >
> > > > > If you have a method which is supposed to perform well, the first example you should try it on is a quadratic. In this case, this should be something like f(x) = x^2 when x>0, (1/2) x^2 when x < 0, since, if the second derivative exists at the optimal solution, one-shot averaging will already give the desired linear speedup as our Theorem 2 shows and there's no need to look at any different method. As for the data points themselves, one can just generate them randomly.
> > > >
> > > > I am not opposed to testing on this data set. It indeed underlines a point in a controlled environment. I also appreciate that the authors experiment under two different noise models. I am more opposed to **only** comparing on synthetic datasets. For instance, in your piecewise quadratic model you introduce additive noise by sampling from a gaussian. How would this noise model compare to the SGD noise, or the noise coming from the data set? It is difficult to make inferences about real-world performance by looking at controlled experiments with a specific choice of parameters. How does one guarantee that these conclusions are robust to the choice of these parameters, and that they are not handpicked? To avoid this I encourage you to compare different algorithms on the a9a/Epsilon data-set as well, which is used by FedAC. And why is FedAC so important? Because it has the best first-order rate that I know of.
> > > >
> > > > > Now IF the method works on a simple quadratic, THEN it makes sense to try it out on real world data sets. If we show that a method does not seem to deliver gains on a quadratic (at least until N=256 nodes), it does not make sense to reply "why didn't you try it on a real world data set?"
> > > >
> > > > Adding on to my previous point, in a controlled synthetic data set it is difficult to know how robust are the conclusions, to different data synthesis choices. For instance, it could be the case that a certain algorithm performs better in a low noise regime. In fact, this is indeed what happens for general convex functions. The mini-max optimal algorithm is a combination of two different algorithms (MB-SGD and single machine SGD), one of which is better based on problem-dependent parameters, such as noise, smoothness, etc. Thus, experimenting on synthetic data sets should not preclude experiments on real datasets (preferably more than one). Thus I think my logic is not backward, in fact, if your proposed algorithm is better, then it should outperform other algorithms on their turf as well!
> > > >
> > > > Furthermore, consider any class of functions $\mathcal{H}$, with two disjoint sub-classes $\mathcal{H}_1$, $\mathcal{H}_2$. You are claiming that if some algorithm doesn't perform well on say $\mathcal{H}_1$, then it doesn't make sense to consider it for any other class within $\mathcal{H}$. But this is wrong, as it could indeed happen that the algorithm performs well on $\mathcal{H}_2$, even if it is not optimal on the entire class $\mathcal{H}$. This is in fact the motivation behind considering assumptions such as quasi-self concordance, etc. In your case, you specifically claim that your method is not for federated learning but well suited for large logistic regression problems, which are convex problems, but not piecewise quadratic problems. I am merely asking you to also evaluate on a logistic regression data set. Please see my comments below for my insistence on equalizing various confounding variables.
> > > >
> > > > And correct me if I am wrong, but on a9a data-set you are concluding that all algorithms perform the same in figure 3. Are you suggesting the experiments in the FedAC paper don't have the correct conclusion then, showing differences in performance of these algorithms? I don't think that is the case, it seems to be a result of considering a single regime of computation-communication.

---

> > > > > ### Author Response · Authors · 2021-08-22
> > > > > **Response 1/2**
> > > > >
> > > > > Thank you for the thoughtful and well-considered response. After this back-and-forth, we believe we are almost entirely in agreement. Please see our response below where we attempt to persuade the reviewer on a few remaining points.
> > > > >
> > > > >
> > > > > > Are you suggesting the experiments in the FedAC paper don't have the correct conclusion then, showing differences in performance of these algorithms?
> > > > >
> > > > >
> > > > > Yes!
> > > > >
> > > > > This is a good discussion to have as it highlights the contribution of our paper. Indeed, all  of the back-and-forth here is focused on Theorem 1. However, our contributions also consist of Theorem 2.
> > > > >
> > > > > According to Theorem 2, one-shot averaging will attain a linear speedup provided  that the **objective is differentiable at the optimal solution**.  With this in mind, let us look at what is being optimized in the FedAC paper.
> > > > >
> > > > > Unfortunately, that paper does not appear  to have a formula for the objective. It does say "The algorithms are tested on l2-regularized logistic regression." It is important for this discussion whether the regularization term is squared or not. We went and looked at the code (link is in the FedAC paper and the relevant bit seems to be on line 139 of the file logistic.py), and the l2 term appears to be added linearly to the gradient, so it must be squared in the objective. The function itself is smooth, and the data set is bounded, so that a Lipschitz constant on the gradient exists.
> > > > >
> > > > > In other words, this is twice-differentiable function. Thus Theorem 2 means that OSA gives you a linear speedup. So if you see anything more than a constant factor speedup, it has to come from not choosing T large enough. So, indeed, we are suggesting the experiments in the FedAC paper have reached the wrong conclusion.
> > > > >
> > > > >
> > > > > Empirically, a version of this phenomenon can be seen in Figure 3 of our paper. If you cut off the simulation early, you might see a difference between the methods. However, this goes away if you simulate long enough. Although the differences in Figure 3 do not appear great even after early cutoff, on some of the other simulations that did not make it to the paper we saw considerable early differences (all to shrink later as a result of Theorem 2).
> > > > >
> > > > > We understand the initial impulse to discount Theorem 2 somewhat, since a version of it was available in the earlier literature under the assumption of a uniform bound on the third derivatives on all of R^n. However:
> > > > >
> > > > > (i) Unlike the earlier literature, our bounds appear to be tight. We prove that twice-differentiability is enough for OSA to get a linear speedup, and in simulations, we show that in the absence of twice differentiability at the optimum point, linear speedup is not achieved.
> > > > >
> > > > > (ii) Our results suggest that, to properly compare methods in this area, they should be tested on strongly convex + smooth objectives which have second derivatives discontinuous at the optimal solution. *This* is the motivation for the simple piecewise quadratic objective. Were the FedAC authors aware of this, we suspect they might have chosen a different objective. As things stood when they wrote their paper, success of OSA depended on a uniform third derivative bound over all of R^n which may/may not hold for their objective.
> > > > >
> > > > > We believe this highlights why showing that OSA gets a linear speedup under the minimal assumption of twice differentiability at the optimal solution is a wortwhile contribution.
> > > > >
> > > > > >  I encourage you to compare different algorithms on the a9a/Epsilon data-set as well, which is used by FedAC
> > > > >
> > > > >
> > > > > We are, of course, happy to include this in the final version of the paper. But our main response on this point is:  we already know what the results will be. This is a consequence of what we wrote just above. If we redo the simulations in the FedAC paper, OSA will deliver a linear speedup with just one communication, because that is what Theorem 2 guarantees.
> > > > >
> > > > > > As reviewer 7aRm pointed out, while you might not want to target Federated Learning as a use case, it is unclear what else is a better use case. And FL is characterized by a large number of user devices, hence my example. In fact, it is not uncommon for the number of machines to run in thousands.
> > > > >
> > > > > Fair enough, but we do not target FL as a use case. The main difference is that in the FL case every device will have its own data, so in FL one really should not claim gradients across nodes are sampled from the same distribution.  Instead, our use case is parallelizing machine learning by giving an identical data set to nodes in the same cluster. Here a typical N seems to be between 10-100, from our sampling of recent papers in parallel ML.
> > > > >
> > > > > > I do understand your motivation to get a rate completely independent of T. But I need to judge the paper based on what was submitted...
> > > > >
> > > > > Not necessarily. We acknowledge that the reviewer's comments will lead to a strong improvement in this paper: before this conversation, we were not even aware of the existence of FedAC. Nevertheless, the reviewer can judge not just on what was submitted, but also our comments here, as well as the promised changes before the final draft -- provided such changes are easy to make, e.g., they do not require developing any new mathematics. We discuss this more below.
> > > > >
> > > > > > This is why I encourage you to experiment with different regimes as in Fig 3./Fig 5. of the FedAC paper. There are important benefits to this kind of comparison... [text omitted] ...I hope it is clear now why I am emphasizing this setup?
> > > > >
> > > > >
> > > > > OK, we agree that these are good arguments. The reviewer is convincing here, and we will include these comparisons in our paper. All the same, we hope the reviewer can acknowledge there are arguments that go the other way as well:
> > > > >
> > > > > (i) Equalizing communications will not show you that one method can have both fewer communications and lower error compared to another.
> > > > >
> > > > > (ii) "#communications to reach error epsilon" and "#iterations to reach error epsilon" under the recommended communication scaling are easy to read off the charts we make for each method, but harder to see from a sequence of charts equalizing the number of communications
> > > > >
> > > > > (iii) The **really important thing** to test is whether the method gives a linear speedup over a reasonable range of n. This is shown clearly in Figure 2 of our paper, but not apparent from repeated comparisons that equalize the number of communications.
> > > > >
> > > > > However, as we have said, the reviewer is quite convincing that there is a lot of value in adding a comparison which equalizes the number of communications, and we will add it  Nevertheless, we next argue  that our existing simulations already make a strong case for the empirical performance of our method  -- though we think the reviewer may already agree  agree with us because they write:
> > > > >
> > > > > > Yes, you are right, and in the experiment, I am suggesting one would still be able to see that advantage, but with a bigger picture around how that advantage changes as more communication becomes feasible
> > > > >
> > > > > **ARE OUR SIMULATIONS FAIR?**
> > > > >
> > > > >
> > > > > While we agree now with the reviewer that the paper would benefit from an additional experiment equalizing the number of communications, we now argue that comparison would only further show the advantages we have already demonstrated. We make two points:
> > > > >
> > > > > (1) Using regularized logistic regression on large-scale data set, as done in the FedAC paper, and as done in Figure 3 of our work, will, by Theorem 2 in this paper, only further show superiority of OSA for this problem.
> > > > >
> > > > > (2) Moving to an example with a discontinuity in the second derivative at the optimal solution:
> > > > >
> > > > > (2a) Synchronous SGD (communicate at every step) always  beats everything else in terms of error/iteration.
> > > > >
> > > > > (2b)  OSA (communicate once at the very end) always beats everything else in terms of error/communication.
> > > > >
> > > > > (2c) Except for those two methods, our method is to the left of every method on the error/iteration chart AND the error/communication chart (FIg (1)a and Fig 1(b)).
> > > > >
> > > > > (2d) Except for OSA, our method communicates *less* than all the other methods we plot.
> > > > >
> > > > > Points (2c) and (2d) are crucial. In the absence of them, we would agree with the reviewer that a comparison could be obscure.  But  together we believe they make a convincing case.
> > > > >
> > > > > Granted, this case was not made very explicitly in our submission, so we understand the reviewer's criticism. But all this is in our paper, it just needs to be explained.  For example, the fact that our method communicates *less* than other methods can be inferred by looking at Figure 1(b) in our paper and observing that the x-axis cuts off earlier for our method.
> > > > >
> > > > > Thus the comparison is not only fair, but arguably somewhat disadvantageous to our method. We will revise our discussion of the experiments to explain this, and we will add the comparison equalizing the number of communications, but these will not contradict the current conclusions. We will also add the baselines suggested by the reviewer [1].
> > > > >
> > > > > Nor will the addition of FedAC revise the conclusions, since, as shown in the graphs in our first response, FedAC appears to perform  poorly on the quadratic example (the version of it with N^{1/3} communications does not even appear to converge here, perhaps due to the difference in the noise model, and we reiterate that the rejoinder that FedAC only communicates three times for N=20 can be countered by pointing out OSA only communicates once).
> > > > >
> > > > > Beyond adding this comparison, we will, of course, significantly revise the discussion of related work and add a polished version of this conversation to a discussion of accelerated FedAC vs unaccelerated Local SGD type methods.

---

> > > > > > ### Comment · Reviewer_3FSZ · 2021-08-29
> > > > > > **Further response**
> > > > > >
> > > > > > > According to Theorem 2, one-shot averaging will attain a linear speedup provided that the objective is differentiable at the optimal solution.
> > > > > >
> > > > > > Yes, your result implies that but only when the noise condition is satisfied, and it is not tautologically true for logistic regression on any dataset?
> > > > > >
> > > > > > > In other words, this is twice-differentiable function. Thus Theorem 2 means that OSA gives you a linear speedup. So if you see anything more than a constant factor speedup, it has to come from not choosing T large enough. So, indeed, we are suggesting the experiments in the FedAC paper have reached the wrong conclusion.
> > > > > >
> > > > > > Again, I don't believe this is true. They have experimented with a large enough T for a9a dataset. I can attest to this because I have worked with this data-set and compared it with FedAC as well. Moreover, as I highlighted above, theorem 2 doesn't imply a linear speedup for OSA for all logistic regression data sets. It does so for the ones, for which the stochastic noise satisfies your assumption. Please note, since you've not addressed it in my comments before, a rate only dependent on T is already available for quadratics, under a more relaxed noise assumption, due to Woodworth et. al. I see theorem 2 as extending this result to logistic regression, under a restrictive noise assumption, which wouldn't necessarily hold true in practice.
> > > > > >
> > > > > > >  If we redo the simulations in the FedAC paper, OSA will deliver a linear speedup with just one communication, because that is what Theorem 2 guarantees.
> > > > > >
> > > > > > Again, I don't see how you are so confident about this. This is not a mathematical truth based on the result itself but depends on the noise.    I agree that it is useful to extend the result of Dieleveut and Patel, Woodworth et. al., but it can not be interpreted as a strict truth in practice. Many practitioners I know of and I, have observed that OSA is not the optimal algorithm to use for logistic regression, in most settings. Thus, I still find this a difficult conclusion to buy. But it is not something you say in the paper, nor it is implied by the theorem itself, thus it is just an over-reading of the theorem itself.
> > > > > >
> > > > > > > Our results suggest that, to properly compare methods in this area, they should be tested on strongly convex + smooth objectives which have second derivatives discontinuous at the optimal solution. This is the motivation for the simple piecewise quadratic objective
> > > > > >
> > > > > > Yes, the synthetic data sets do make sense in light of the result.
> > > > > >
> > > > > > > The main difference is that in the FL case every device will have its own data, so in FL one really should not claim gradients across nodes are sampled from the same distribution.  Instead, our use case is parallelizing machine learning by giving an identical data set to nodes in the same cluster.
> > > > > >
> > > > > > That's not entirely true. Cross-silo FL (Kairouz et. al.) is in fact a realistic setting, where data between the workers could be somewhat similar, such as human x-ray in multiple hospitals. Furthermore, in reality, many settings have low heterogeneity and one hopes that the implications of the homogeneous setting still hold (c.f. Woodworth et. al. "Minibatch vs Local SGD for Heterogeneous Distributed Learning"). Having said that if you claim you are in the intra-cluster setting, with the same finite data given to different machines, why would you not use distributed SVRG/SAG? It is unclear to me if the FL connection can be skimmed away.
> > > > > >
> > > > > > > However, as we have said, the reviewer is quite convincing that there is a lot of value in adding a comparison which equalizes the number of communications, and we will add it
> > > > > >
> > > > > > Ok, I think we are on the same page about this then.
> > > > > >
> > > > > > > Theorem 2 on one-shot averaging has a crisp message. In particular, as we argued above, it answers the reviewer's question about the simulations in FedAC. It also shows that re-doing the large-scale simulation in FedAC, as suggested by the reviewer, will not lead to anything different than what is already shown in Figure 3.
> > > > > >
> > > > > > I hope it is clear why in the light of the discussion about the noise condition this is an underqualified statement to make?
> > > > > >
> > > > > > >  Though wouldn't "single machine SGD" suggested by the reviewer either have the same performance as what we call Synchronous SGD, or exactly factor of N worse, depending on how the terms are defined? It may be that we do not have the same definitions in mind as the reviewer here.
> > > > > >
> > > > > > Yes, it should have factor N worse performance. But I am asking you to compare against the combination of single machine SGD + large mini-batch SGD. The motivation can be found in the paper by Woodworth et. al. : "The Min-Max Complexity of Distributed Stochastic Convex Optimization with Intermittent Communication".

---

> > > > > > > ### Author Response · Authors · 2021-08-30
> > > > > > > **Simulations**
> > > > > > >
> > > > > > > We went ahead and forked the github from FedAC paper and added the code for our method in order to do a comparison. The results can be seen at:
> > > > > > >
> > > > > > > https://anonymous.4open.science/r/FedAc-NeurIPS20-3B6F/README.md
> > > > > > >
> > > > > > > https://anonymous.4open.science/r/FedAc-NeurIPS20-3B6F/browse_figures.ipynb
> > > > > > >
> > > > > > > The results are consistent with what we have written earlier: given that our method outperformed at the recommended communication level while **doing fewer communications** than the methods we compared to, it continues to outperform once the number of communication is equalized.
> > > > > > >
> > > > > > > In more detail:
> > > > > > >
> > > > > > > -- our method is the curve labeled "Linear" (since we are linearly growing the inter-communication intervals)
> > > > > > >
> > > > > > > -- FedAvg is local gradient descent, and we also plot minibatch SGD and accelerated minibatch SGD.
> > > > > > >
> > > > > > > -- We use the same cost function/data set as Figures 1/3 of the FedAC paper (binary logistic regression + l_2 regularization), and similarly we plot T=4096 iterations.
> > > > > > >
> > > > > > > -- We made a few changes to make save computation time.
> > > > > > >
> > > > > > > First, we used step-size tuning from  {0.001, 0.003, 0.01, 0.03, 0.1, 0.3, 1, 3} , which is a more granular set than what is in the FedAC paper. We used this for all methods except ours, for which we just used the theoretical step-size 3/lambda*(iter+1); for comparison, our step-size goes from 1.5 to 0.001 over this range.
> > > > > > >
> > > > > > > Second, we used lambda = 1.0 (l_2 reg. factor or strong convexity) in contrast to the FedAC paper which uses a much smaller lambda of 10^{-3}; this is because higher condition numbers require more computation time and we wanted to post this response before the discussion period ran out. We will also experiment with smaller lambdas and post the results here if they finish before the end of the discussion period.
> > > > > > >
> > > > > > > -- For reasons not entirely clear to us, the process of making the github anonymous has doubled every line (so the code is repeated twice).
> > > > > > >
> > > > > > > The graphs clearly show the better performance of our method -- in spite of the fact that the comparison disadvantages our method, as we tune step-sizes for our method but not our own. We believe this answers the reviewer's main objections: (i) we are now equalizing the number of communications (ii) the data sets are the same real data sets as in the FedAC paper.
> > > > > > >
> > > > > > >
> > > > > > > > Yes, your result implies that but only when the noise condition is satisfied, and it is not tautologically true for logistic regression on any dataset?
> > > > > > >
> > > > > > > > theorem 2 doesn't imply a linear speedup for OSA for all logistic regression data sets. It does so for the ones, for which the stochastic noise satisfies your assumption.
> > > > > > >
> > > > > > > > This is not a mathematical truth based on the result itself but depends on the noise.
> > > > > > >
> > > > > > > Our assumption is that noise is subgaussian, which is quite a mild assumption.
> > > > > > >
> > > > > > > However, our main response is: **if you look at the anonymous github above, we did the simulation.** Specifically, if you look at the graph labeled K=4096, "Linear" is just is one-shot averaging (because T=4096). As we asserted in our earlier response, it performs as well as communicating more (compare the error to the other graphs by looking at the y-axis) in spite of doing only a single communication.
> > > > > > >
> > > > > > > > Having said that if you claim you are in the intra-cluster setting, with the same finite data given to different machines, why would you not use distributed SVRG/SAG?
> > > > > > >
> > > > > > > The standard "linear rate" results SAGA/SVRG do not work under the assumptions in this paper: they assume that the noise comes from  sampling data points. Our result here (i.e., Theorem 1) applies to that setting, but it also works for arbitrary zero mean noise.
> > > > > > >
> > > > > > >
> > > > > > > > Please note, since you've not addressed it in my comments before, a rate only dependent on T is already available for quadratics, under a more relaxed noise assumption, due to Woodworth et. al. I see theorem 2 as extending this result to logistic regression, under a restrictive noise assumption, which wouldn't necessarily hold true in practice.
> > > > > > >
> > > > > > > Well, the contributions of Theorem 2 are stronger than that: it is not limited to logistic regression. In general, obtaining this result for convex functions is considerably more difficult than obtaining it for quadratics.  And if we did not say this explicitly before, we will certainly look at all the references brought to our attention and make sure their contributions are discussed in the final version of the paper.

---

> > > > > > > > ### Author Response · Authors · 2021-08-30
> > > > > > > > **Typo**
> > > > > > > >
> > > > > > > > > ...as we tune step-sizes for our method but not our own...
> > > > > > > >
> > > > > > > > This should be: "...as we tune step-sizes for _other_ _methods_ but not our own...."

---

> > > > > ### Author Response · Authors · 2021-08-22
> > > > > **Response 2/2**
> > > > >
> > > > > *Summary and conclusion*:
> > > > >
> > > > > We want to thank the reviewer for bringing the FedAC paper, of which we were unaware, to our attention. We would also like to thank the reviewer for being willing to revise their score. We have learned a lot from this interaction, which could only have happened within the anonymous framework of openreview, as the usual interactions among scientists are too "polite" and one does not always get the necessary feedback to improve and sharpen one's claims.
> > > > >
> > > > >
> > > > > Our response, though somewhat lengthy, really boils down to two main points:
> > > > >
> > > > > (i) Theorem 2 on one-shot averaging has a crisp message. In particular, as we argued above, it answers the reviewer's question about the simulations in FedAC. It also shows that re-doing the large-scale simulation in FedAC, as suggested by the reviewer, will not lead to anything different than what is already shown in Figure 3.
> > > > >
> > > > > (iii) Our comparisons show that, at the recommended communication settings, our method achieves lower error while also communicating less. While fully acknowledging the reviewer's point on the benefits of adding charts equalizing the number of communications, this is also a meaningful thing to show. In particular, the comparison is fair.
> > > > >
> > > > > We believe these are new points, i.e., we are not simply regurgitating what was said in earlier rounds but bringing something additional to the conversation. In light of all this, we wonder if we could ask the reviewer to reconsider their rating further.
> > > > >
> > > > > [1] Though wouldn't "single machine SGD" suggested by the reviewer either have the same performance as what we call Synchronous SGD, or exactly factor of N worse, depending on how the terms are defined? It may be that we do not have the same definitions in mind as the reviewer here. Also, SCAFFOLD was developed for the heterogeneous case, which is why we did not originally include it (we only compared with the literature that assumes gradients are sampled from the same distribution), but we will include it in the final version.

---

> > > > ### Comment · Reviewer_3FSZ · 2021-08-19
> > > > **Response to authors 2/2**
> > > >
> > > > > There's nothing difficult about it. One can plot, as we do, error per iteration and error per communication in two different graphs. This should give you a complete picture of how well the method performs on two different metrics.
> > > >
> > > > Communication-computation trade-offs don't exist in a vacuum. There are settings where communication is indeed the bottleneck such as synchronous cross-device federated learning. But there are also settings where it is not really a bottleneck, such as a data center with multiple nodes connected through high-speed links. And there are all scenarios in between, where communication v/s computation costs differently influence the true wall clock time. Convergence theory shouldn't develop oblivious to this, where one tests their algorithms in the regimes where they perform well. This is why I encourage you to experiment with different regimes as in Fig 3./Fig 5. of the FedAC paper. There are important benefits to this kind of comparison:
> > > >
> > > > 1) It is difficult to know which regime/setting one is operating in, and what is the actual physical time per communication. Thus, by equalizing the rounds of communication between different algorithms one equalizes this cost.
> > > >
> > > > 2) With the communication cost fixed if the algorithms also have access to a fixed computation budget (such as flops or first-order oracle access) then one can extrapolate the wall clock times of each of these algorithms.
> > > >
> > > > 3) In some settings it also makes sense to further equalize the computation per communication round for balancing the load over time.
> > > >
> > > > 3 can't be achieved when the local steps grow linearly with time (as in your case), but it is perfectly reasonable to expect 1 and 2. Note that this doesn't inhibit anything. In such a comparison, algorithms which are good in a sparse-communication regime would be easily apparent, as is the case in Fig 3./Fig 5. of the FedAC paper. **I hope it is clear now why I am emphasizing this setup?**
> > > >
> > > > Moreover, as I already said before, I would like to see other important baselines such as large mini-batch SGD and single machine SGD, etc (which match the lower bound for general convex functions in certain settings).
> > > >
> > > > > Contrary to what the reviewer writes, there's no difficulty comparing methods with different #communications. Indeed, consider one-shot averaging vs synchronous SGD on a smooth problem, shown in Figure 3b in our paper. From Theorem 2 we know they should be roughly the same, and indeed we see the performance per iteration is nearly identical...even though the former does 1 communication and the latter does N communications. We immediately conclude the former is better (similar error with fewer communications).
> > > >
> > > > Yes, you are right, and in the experiment, I am suggesting one would still be able to see that advantage, but with a bigger picture around how that advantage changes as more communication becomes feasible. For instance, in figure 3 in the FedAC paper, one-shot averaging would appear as a special case of FedAVG on the far right of each sub-plot. Based on your assertion, either the figure in the FedAC is wrong because it shows different performances? Or you are claiming that all the algorithms perform very comparable to each other and the performances are magnified here? If that is the case then perhaps choosing a data-set like CovType/epsilon makes sense?
> > > >
> > > > I am also curious now how the step-sizes were tuned in your experiments? Were they tuned for best final loss/best overall loss/best loss averaged over the final iterations?
> > > >
> > > > > Our main takeaway from this discussion is that the reviewer is rather biased in favor of FedAC. The response that a step-size which depends on the distance to the optimal solution does not actually depend on it but just requires "step size tuning" is particularly puzzling and troublesome.
> > > >
> > > > I can assure you that I am not biased in favor of FedAC. It is an already published work, with the best first-order analysis, and to show and discuss improvements upon it, is necessary for any new work. I have clarified the step-size issue above. It is definitely a limitation of FedAC's theory, but not detrimental in my opinion. And, it is specially not an excuse for the fact that they do have a significant improvement in terms of N.
> > > >
> > > > **Comparison to the existing rate on Quadratics**
> > > >
> > > > It is already known due to Woodworth et. al. that accelerated FedAvg with any rounds of communication, including OSA, is minimax optimal for quadratic functions. In other words, the upper bound depends on $T$, and not $R$ or $K$ alone. How does the result in Theorem 2, compare to this? It seems to me (as I also wrote in my original review) that it is a generalization of that result. This is an important point to discuss in the paper. As you point out in your experiments with logistic regression, it follows the assumptions of Theorem 2 except for the noise condition. If the noise condition were not restrictive, theorem 2 would imply that on logistic regression, OSA should be as good as MB-SGD or FedAvg with more communication rounds for the same number of iterations. This is one of the reasons, why I am insisting on more experiments with logistic regression: to understand how crucial is the sub-gaussian noise assumption.
> > > >
> > > > **Summary**
> > > >
> > > > To summarize I think we are on the same page regarding the theoretical contribution. I would encourage the authors to add a detailed comparison to FedAC which goes into the nuances of where FedAC is better than their algorithm, and in what respects it is not. What I am still not sure of is the empirical benefit of OSA or linearly decreasing communication frequency against all the baselines, in a fair setup, i.e., equalizing for communication as well as total computation. In light of this discussion and under the assumption that the authors will address the comparison with FedAC, I will improve my score to a 5.  I am open to further increasing it if my concerns regarding the experiments are resolved.
> > > >
> > > > **Reference**
> > > >
> > > > Woodworth, Blake, et al. "Is local SGD better than minibatch SGD?." International Conference on Machine Learning. PMLR, 2020.

---

### Decision · Program_Chairs · 2021-09-27

**Decision:**

Accept (Poster)

**Comment:**

This paper studies the oracle (gradient) complexity of the local SGD algorithm (and one-shot averaging as a special case). The paper makes two main contributions (see additional comments below), (1) a slightly improved complexity estimate for local SGD with non-equally spaced (linearly increasing) intermissions between communication rounds, and (2) a novel asymptotic result for one-shot averaging under PL and second order smoothness assumption.

The paper has prompted lengthy discussions, both publicly and internally among the reviewers.

The main focus of the paper is currently on result (1). However, all reviewers unanimously find this result not very significant, since, for example a discussion of a corresponding lower complexity bound is missing and thus a log factor improvement seems very marginal. For the final version, we urge the authors
- to carefully discuss the improvement over Theorem 2 (strongly convex case) in [[Woodworth et al, ICML 2020](https://arxiv.org/pdf/2002.07839.pdf)] (essentially, it seems the new result improves under the condition $\kappa \leq \log (T)$ only.
- the reviewers mentioned the suboptimal dependence on e.g. the condition number and found the author's response (speculating on an inherent trade-off) very interesting. However, I would strongly encourage authors to be very precise when including such claims, as the trade-off may just be a consequence of the proof technique.

The reviewers found result (2) interesting. Although strong assumptions are being made at the moment, the result could possibly stipulate further research.

To conclude, the contribution regarding (2), as well as the proposed modifications in the discussion with the reviewers (including discussion on FedAC and additional numerical results), gave the impetus to argue for acceptance for this work.